# Somatic mutation rates scale with lifespan across mammals

Alex Cagan[1,15] ✉, Adrian Baez-Ortega[1,15], Natalia Brzozowska[1], Federico Abascal[1], Tim H. H. Coorens[1], Mathijs A. Sanders[1,2], Andrew R. J. Lawson[1], Luke M. R. Harvey[1], Shriram Bhosle[1], David Jones[1], Raul E. Alcantara[1], Timothy M. Butler[1], Yvette Hooks[1], Kirsty Roberts[1], Elizabeth Anderson[1], Sharna Lunn[1], Edmund Flach[3], Simon Spiro[3], Inez Januszczak[3,4], Ethan Wrigglesworth[3], Hannah Jenkins[3], Tilly Dallas[3], Nic Masters[3], Matthew W. Perkins[5], Robert Deaville[5], Megan Druce[6,7], Ruzhica Bogeska[6,7], Michael D. Milsom[6,7], Björn Neumann[8,9], Frank Gorman[10], Fernando Constantino-Casas[10], Laura Peachey[10,11], Diana Bochynska[10,12], Ewan St. John Smith[13], Moritz Gerstung[14], Peter J. Campbell[1], Elizabeth P. Murchison[10], Michael R. Stratton[1] & Iñigo Martincorena[1] ✉

The rates and patterns of somatic mutation in normal tissues are largely unknown outside of humans[1–7]. Comparative analyses can shed light on the diversity of mutagenesis across species, and on long-standing hypotheses about the evolution of somatic mutation rates and their role in cancer and ageing. Here we performed whole-genome sequencing of 208 intestinal crypts from 56 individuals to study the landscape of somatic mutation across 16 mammalian species. We found that somatic mutagenesis was dominated by seemingly endogenous mutational processes in all species, including 5-methylcytosine deamination and oxidative damage. With some differences, mutational signatures in other species resembled those described in humans[8], although the relative contribution of each signature varied across species. Notably, the somatic mutation rate per year varied greatly across species and exhibited a strong inverse relationship with species lifespan, with no other life-history trait studied showing a comparable association. Despite widely different life histories among the species we examined—including variation of around 30-fold in lifespan and around 40,000-fold in body mass—the somatic mutation burden at the end of lifespan varied only by a factor of around 3. These data unveil common mutational processes across mammals, and suggest that somatic mutation rates are evolutionarily constrained and may be a contributing factor in ageing.

Somatic mutations accumulate in healthy cells throughout life. They underpin the development of cancer[9] and, for decades, have been speculated to contribute to ageing[10–12]. Directly studying somatic mutations in normal tissues has been challenging owing to the difficulty of detecting mutations present in single cells or small clones in a tissue. Only recent technological developments, such as in vitro expansion of single cells into colonies[13,14], microdissection of histological units[8,15], single-cell sequencing[16,17] or single-molecule sequencing[18], are beginning to enable the study of somatic mutation in normal tissues.

Over the last few years, studies in humans have started to provide a detailed understanding of somatic mutation rates and the contribution of endogenous and exogenous mutational processes across normal tissues[8,13,14,19,20]. These studies are also revealing how, as we age, some human tissues are colonized by mutant cells that contain cancer-driving mutations, and how this clonal composition changes with age and disease. With the exception of some initial studies, far less is known about somatic mutation in other species[1–7]. Yet, comparative analyses of somatic mutagenesis would shed light on the diversity of mutagenic processes across species, and on long-standing questions regarding the evolution of somatic mutation rates and their role in cancer and ageing.

A decades-long hypothesis on the evolution of somatic mutation rates pertains to the relationship between body mass and cancer risk. Some models predict that the risk of cancer should increase proportionally to the number of cells at risk of transformation. However, there appears to be no correlation between body mass and cancer risk across

[1]Cancer, Ageing and Somatic Mutation (CASM), Wellcome Sanger Institute, Hinxton, UK. [2]Department of Hematology, Erasmus MC Cancer Institute, Rotterdam, the Netherlands. [3]Wildlife Health Services, Zoological Society of London, London, UK. [4]The Natural History Museum, London, UK. [5]Institute of Zoology, Zoological Society of London, London, UK. [6]Division of Experimental Hematology, German Cancer Research Center (DKFZ), Heidelberg, Germany. [7]Heidelberg Institute for Stem Cell Technology and Experimental Medicine GmbH (HI-STEM), Heidelberg, Germany. [8]Wellcome Trust–Medical Research Council Cambridge Stem Cell Institute, University of Cambridge, Cambridge, UK. [9]Department of Clinical Neurosciences, University of Cambridge, Cambridge, UK. [10]Department of Veterinary Medicine, University of Cambridge, Cambridge, UK. [11]Bristol Veterinary School, Faculty of Health Sciences, University of Bristol, Langford, UK. [12]Department of Pathology, Faculty of Veterinary Medicine, Universitatea de Stiinte Agricole si Medicina Veterinara, Cluj-Napoca, Romania. [13]Department of Pharmacology, University of Cambridge, Cambridge, UK. [14]European Molecular Biology Laboratory, European Bioinformatics Institute (EMBL-EBI), Hinxton, UK. [15]These authors contributed equally: Alex Cagan, Adrian Baez-Ortega. ✉e-mail: ac36@sanger.ac.uk; im3@sanger.ac.uk

species[21,22]. This observation, known as Peto's paradox, suggests that the evolution of larger body sizes is likely to require the evolution of stronger cancer suppression mechanisms[23,24]. Whether evolutionary reduction of cancer risk across species is partly achieved by a reduction of somatic mutation rates remains unknown.

A second long-standing hypothesis on the evolution of somatic mutation rates relates to the proposed role of somatic mutations in ageing. Multiple forms of molecular damage, including somatic mutations, telomere attrition, epigenetic drift and loss of proteostasis, have been proposed to contribute to ageing, but their causal roles and relative contributions remain debated[25,26]. Evolutionary theory predicts that species will evolve protection or repair mechanisms against life-threatening damage to minimize death from intrinsic causes, but that selection is too weak to delay ageing far beyond the typical life expectancy of an organism in the wild (Supplementary Note 1). If somatic mutations contribute to ageing, theory predicts that somatic mutation rates may inversely correlate with lifespan across species[27,28]. This prediction has remained largely untested owing to the difficulty of measuring somatic mutation rates across species.

## Detection of somatic mutations across species

The study of somatic mutations with standard whole-genome sequencing requires isolating clonal groups of cells recently derived from a single cell[8,13,14]. To study somatic mutations across a diverse set of mammals, we isolated 208 individual intestinal crypts from 56 individuals across 16 species with a wide range of lifespans and body sizes: black-and-white colobus monkey, cat, cow, dog, ferret, giraffe, harbour porpoise, horse, human, lion, mouse, naked mole-rat, rabbit, rat, ring-tailed lemur and tiger (Supplementary Table 1). We chose intestinal crypts for several reasons. First, they are histologically identifiable units that line the epithelium of the colon and small intestine and are amenable to laser microdissection. Second, human studies have confirmed that individual crypts become clonally derived from a single stem cell and show a linear accumulation of mutations with age, which enables the estimation of somatic mutation rates through genome sequencing of single crypts[8]. Third, in most human crypts, most somatic mutations are caused by endogenous mutational processes common to other tissues, rather than by environmental mutagens[8,18].

A colon sample was collected from each individual, with the exception of a ferret from which only a small intestine sample was available. This sample was included because results in humans have shown that the mutation rates of colorectal and small intestine epithelial stem cells are similar[14,20] (Extended Data Fig. 1). We then used laser microdissection on histological sections to isolate individual crypts for whole-genome sequencing with a low-input library preparation method[29] (Fig. 1a, Extended Data Fig. 2, Supplementary Table 2), with the exception of human crypts, for which sequencing data were obtained from a previous study[8]. A bioinformatic pipeline was developed to call somatic mutations robustly in all these species despite the variable quality of their genome assemblies (Methods). The distribution of variant allele fractions of the mutations detected in each crypt confirmed that crypts are clonal units in all species, enabling the study of somatic mutation rates and signatures (Extended Data Fig. 3).

We found substantial variation in the number of somatic single-base substitutions across species and across individuals within each species (Fig. 1b). For five species with samples from multiple individuals (dog, human, mouse, naked mole-rat and rat), linear regression confirmed a clear accumulation of somatic mutations with age (Fig. 1c, Extended Data Fig. 4, Supplementary Table 3). All linear regressions were also consistent with a non-significant intercept. This resembles observations in humans[20] and suggests that the time required for a single stem cell to drift to fixation within a crypt is a small fraction of the lifespan of a species. This facilitates the estimation of somatic mutation rates across species by dividing the number of mutations in a crypt by the age of the individual (Supplementary Table 4). The number of somatic insertions and deletions (indels) was consistently lower than that of substitutions in all crypts (Fig. 1b), in agreement with previous findings in humans[8].

## Mutational signatures across mammals

Somatic mutations can be caused by multiple mutational processes, involving different forms of DNA damage and repair. Different processes cause characteristic frequencies of base substitution types and indels at different sequence contexts, often referred to as mutational signatures, which can be inferred from mutation data[30]. Across species, the mutational spectra showed clear similarities, with a dominance of cytosine-to-thymine (C>T) substitutions at CpG sites, as observed in human colon, but with considerable variation in the frequency of other substitution types (Fig. 2a). To quantify the contribution of different mutational processes to the observed spectra, we applied mutational signature decomposition[8,30]. We used a Bayesian model to infer mutational signatures de novo, while accounting for differences in genome sequence composition across species, and using the COSMIC human signature SBS1 (C>T substitutions at CpG sites) as a fixed prior to ensure its complete deconvolution[31] (Methods). This approach identified two signatures beyond SBS1, labelled SBSB and SBSC, which resemble COSMIC human signatures SBS5 and SBS18, respectively (cosine similarities 0.93 and 0.91) (Fig. 2b).

This analysis suggests that the same three signatures that dominate somatic mutagenesis in the human colon are dominant in other mammals: SBS1, which is believed to result from the spontaneous deamination of 5-methylcytosine[8,32]; SBSB (SBS5), a common signature across human tissues that may result from endogenous damage and repair[18,33]; and SBSC (SBS18), which is dominated by C>A substitutions and attributed to oxidative DNA damage[30]. Signature SBSC contains a minor component of T>A substitutions (resembling COSMIC SBS34), which appear to be the result of DNA polymerase slippage at the boundaries between adjacent adenine and thymine homopolymer tracts, but could also reflect assembly errors at those sites[33]. Although all of the species that we examined shared the three mutational signatures, their contributions varied substantially across species (Fig. 2c). SBSC was particularly prominent in mouse and ferret, and the ratio of SBS1 to SBSB/5 varied from approximately 1.2 in rat or rabbit to 6.4 in tiger. In several species with data from multiple individuals, separate linear regressions for each signature confirmed that mutations from all three signatures accumulate with age (Fig. 2d, Extended Data Fig. 5).

Although signature deconvolution identified three signatures that are active across species, we noticed some differences in the mutational profile of signature SBSB among species. To investigate this further, we inferred independent versions of SBSB from each species, while accounting for differences in genome sequence composition (Methods). This revealed inter-species variability in the mutational profile of this signature, particularly in the C>T component (Extended Data Fig. 6). Species-specific versions of SBSB showed different similarities to the related human signatures SBS5 and SBS40. For example, SBSB inferred from the human data showed a stronger similarity with the reference human signature SBS5 (cosine similarities with SBS5 and SBS40: 0.93 and 0.84), whereas SBSB from rabbit more closely resembled the reference human signature SBS40 (0.87 and 0.91). These observations are consistent with the hypothesis that SBS5 and SBS40 result from a combination of correlated mutational processes, with some variation across human tissues[18,33] and across species.

Analysis of the indel mutational spectra revealed a dominance of the human indel signatures ID1 and ID2, which are characterized by single-nucleotide indels at A/T homopolymers, and probably caused by strand slippage during DNA replication[30] (Extended Data Fig. 7a). The ratio of insertions (ID1) to deletions (ID2) appears to vary across species, possibly reflecting a differential propensity for slippage of the template and nascent DNA strands[30]. In addition, the indel spectra suggest a potential contribution of signature ID9 (the aetiology of which

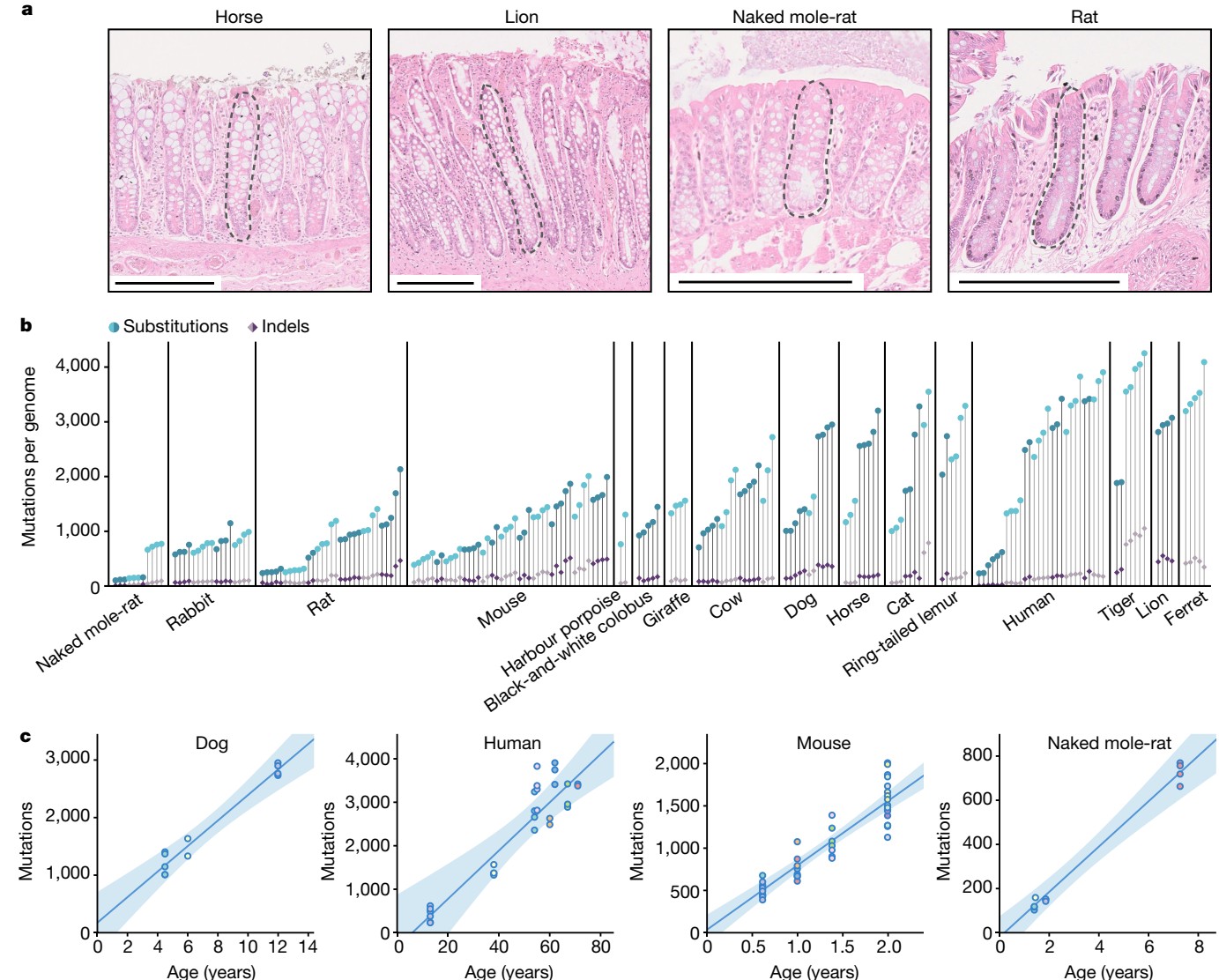

**Fig. 1 | Somatic mutation burden in mammalian colorectal crypts. a,** Histology images of colon samples from horse, lion, naked mole-rat and rat, with one colorectal crypt marked in each. Scale bars, 250 μm. **b,** Burden of somatic substitutions and indels per diploid genome in each colorectal crypt sample (corrected for the size of the analysable genome). Samples are grouped by individual, with samples from the same individual coloured in the same shade. Species, and individuals within each species, are sorted by mean mutation burden. **c,** Linear regression of somatic substitution burden (corrected for analysable genome size) on individual age for dog, human, mouse and naked mole-rat samples. Samples from the same individual are shown in the same colour. Regression was performed using mean mutation burdens per individual. Shaded areas indicate 95% confidence intervals of the regression line.

remains unknown) to human, colobus, cow, giraffe and rabbit. Analysis of indels longer than one base pair also suggested the presence of a signature of four-base-pair insertions at tetrameric repeats, which was particularly prevalent in mouse and tiger; a pattern of insertions of five or more base pairs at repeats in colobus; and a pattern of deletions of five or more base pairs at repeats, which was prominent in rabbit and resembles ID8 (a signature possibly caused by double-strand break repair through non-homologous end joining[30]) (Extended Data Fig. 7a).

## Other mutational processes and selection

The apparent lack of additional mutational signatures is noteworthy. A previous study of 445 colorectal crypts from 42 human donors found that many crypts were affected by a signature that was later attributed to colibactin, a genotoxin produced by *pks*+ strains of *Escherichia coli*[8,34,35]. Analysing the original human data and our non-human data with the same methodology, we found evidence of colibactin

mutagenesis in 21% of human crypts, but only uncertain evidence of colibactin in one non-human crypt (0.6%) (Extended Data Fig. 7b, Methods). This revealed a significant depletion of colibactin mutagenesis in the non-human crypts studied (Fisher's exact test, $P = 7 \times 10^{-14}$). The apparent difference in colibactin mutagenesis observed between species, or between the cohorts studied, might result from a different prevalence of *pks*+ *E. coli* strains[36] or a different expression of colibactin by *pks*+ *E. coli* across species[37]. Finally, we also searched for evidence of APOBEC signatures (SBS2 and SBS13), which have been reported in a small number of human crypts and are believed to be caused by APOBEC DNA-editing cytidine deaminases. We detected APOBEC signatures in 2% (*n* = 9) of human crypts and found only uncertain evidence in one non-human crypt (*P* = 0.30).

Beyond substitutions and indels, crypts from the eight species with chromosome-level genome assemblies were inspected for large-scale copy number changes (at least 1 Mb) (Methods). Studies in humans have found that large-scale copy number changes are relatively rare in

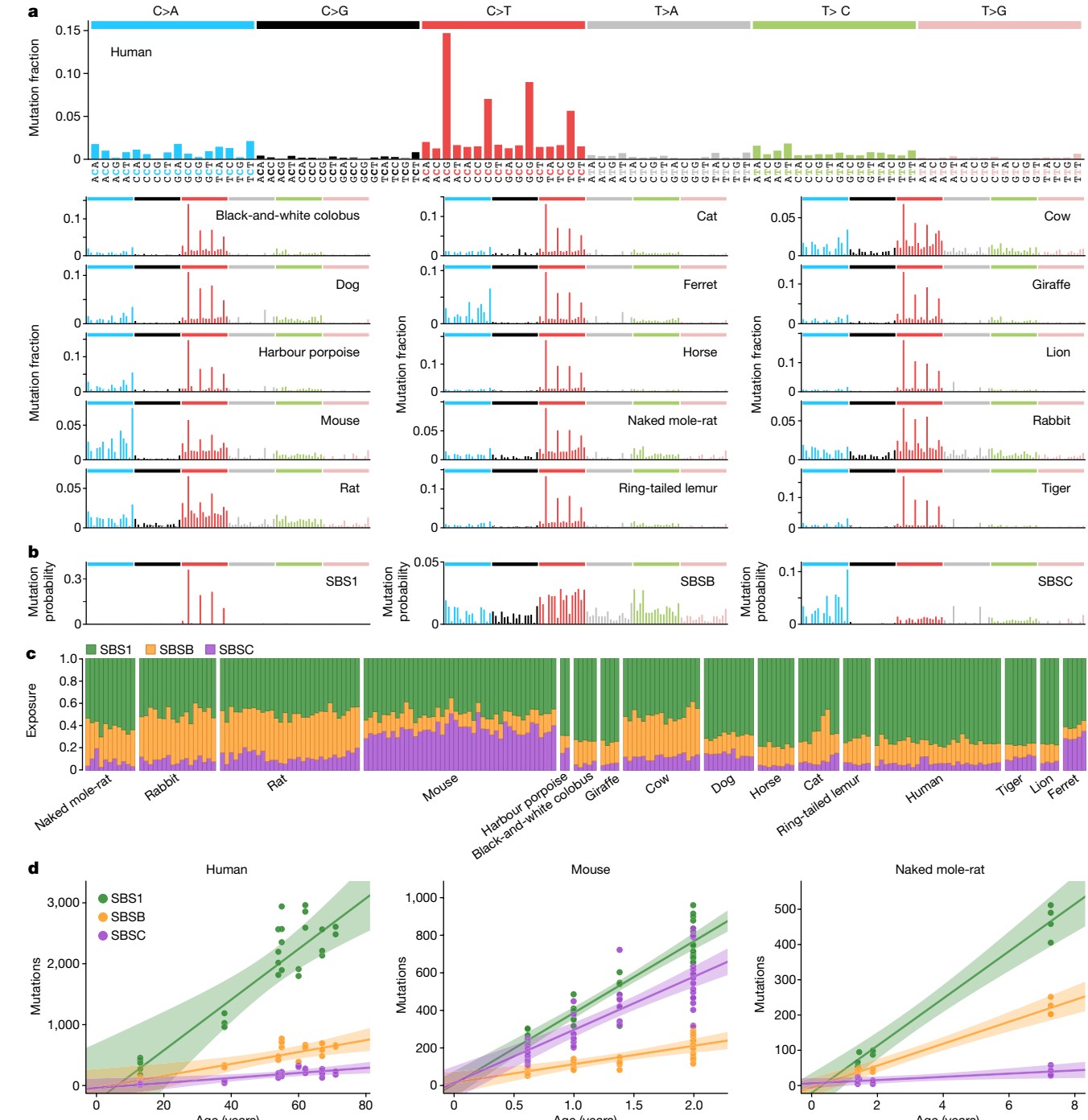

**Fig. 2 | Mutational processes in the mammalian colon. a**, Mutational spectra of somatic substitutions in each species. The *x* axis shows 96 mutation types on a trinucleotide context, coloured by base substitution type. **b**, Mutational signatures inferred from (SBSB, SBSC) or fitted to (SBS1) the species mutational spectra shown in **a**, and normalized to the human genome trinucleotide frequencies. The *y* axis shows mutation probability. **c**, Estimated contribution of each signature to each sample. Samples are arranged horizontally as in Fig. 1b. **d**, Linear regression of signature-specific mutation burdens (corrected for analysable genome size) on individual age for human, mouse and naked mole-rat samples. Regression was performed using mean mutation burdens per individual. Shaded areas indicate 95% confidence intervals of the regression line.

normal tissues, including colorectal epithelium[8]. Consistent with these results, we only identified 4 large copy number changes across the 162 crypts included in this analysis: 2 megabase-scale deletions in 2 crypts from the same cow; the loss of an X chromosome in a female mouse crypt; and a 52-Mb segment with copy-neutral loss of heterozygosity in a human crypt (Extended Data Fig. 8, Methods). These results suggest that large-scale somatic copy number changes in normal tissues are also rare in other mammalian species.

Previous analyses in humans have shown that most somatic mutations in colorectal crypts accumulate neutrally, without clear evidence of negative selection against non-synonymous mutations and with a low frequency of positively selected cancer-driver mutations[8]. To study somatic selection in our data, we calculated the exome-wide ratio of non-synonymous to synonymous substitution rates (dN/dS) in each of the 12 species with available genome annotation. To do so and to detect genes under positive selection, while accounting for the effects of trinucleotide

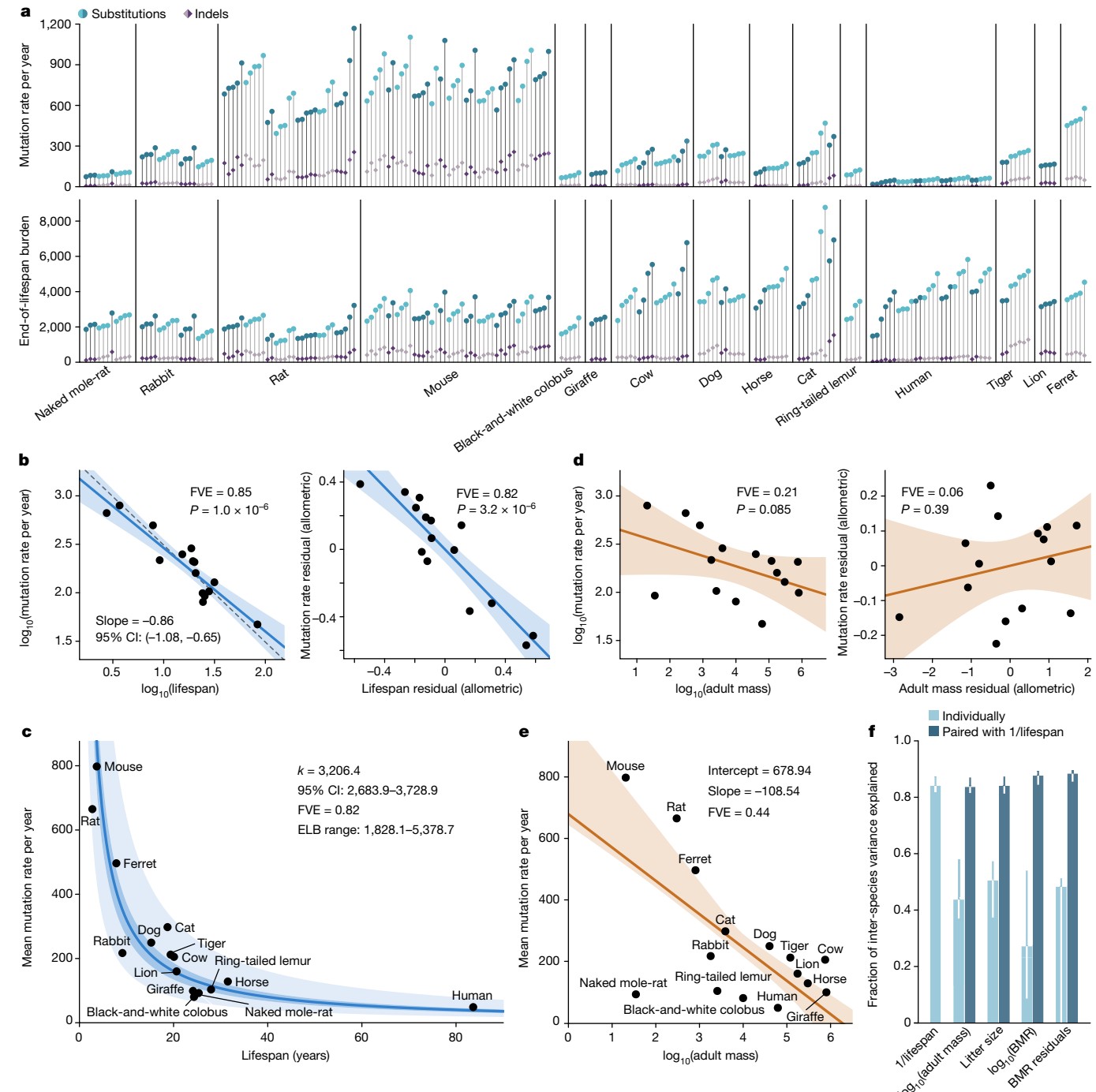

**Fig. 3 | Associations between somatic mutation rates and life-history traits.**
**a**, Somatic mutation rate per year and expected end-of-lifespan mutation burden (ELB) per crypt. Samples are arranged horizontally as in Fig. 1b; harbour porpoise samples were excluded owing to the age of the sampled individual being unknown. **b**, Left, allometric regression of somatic mutation rate on lifespan. Right, regression of body-mass-adjusted residuals for somatic mutation rate and lifespan (partial correlation; Methods). Regression was performed using mean mutation rates per species. Shaded areas represent 95% confidence intervals (CI) of regression lines. FVE and $P$ values (by $F$-test) are indicated (note that, for simple linear regression, FVE = $R^2$). The dashed line denotes a reference slope of −1. **c**, Zero-intercept LME regression of somatic mutation rate on inverse lifespan (1/lifespan), presented on the scale of untransformed lifespan ($x$ axis). For simplicity, the $y$ axis shows mean mutation rates per species, although rates per crypt were used in the regression. The

darker shaded area indicates 95% CI of the regression line, and the lighter shaded area marks a twofold deviation from the line. Point estimate and 95% CI of the regression slope ($k$), FVE and range of end-of-lifespan burden are indicated. **d**, Allometric regression and linear regression of lifespan-adjusted residuals, for somatic mutation rate and body mass (elements as described in **b**). **e**, Free-intercept LME regression of somatic mutation rate on log-transformed body mass. The $y$ axis shows mean mutation rates per species, although rates per crypt were used in the regression. Shaded area indicates 95% bootstrap interval of the regression line ($n$ = 10,000 replicates). Point estimates of the regression intercept and slope, and FVE, are indicated. **f**, FVE values for free-intercept LME models using 1/lifespan or other life-history variables (alone or combined with 1/lifespan) as explanatory variables. Error bars indicate 95% bootstrap intervals ($n$ = 10,000).

**Table 1 | Variation in adult body mass, lifespan, somatic mutation rate and end-of-lifespan mutation burden across the 16 mammalian species surveyed**

| Variable | Minimum | Maximum | Fold variation |
|---|---|---|---|
| **Adult mass (g)** | 20.50 | 800,000.00 | 39,024.39 |
| **Lifespan (years)** | 2.75 | 83.67 | 30.44 |
| **Mutation rate per year (substitutions per genome)** | 47.12 | 796.42 | 16.90 |
| **End-of-lifespan burden (substitutions per genome)** | 1,828.08 | 5,378.73 | 2.94 |

Species-level estimates are provided in Supplementary Tables 3 and 6.

sequence context and mutation rate variation across genes, we used the dNdScv model[38] (Methods). Although the limited number of coding somatic mutations observed in most species precluded an in-depth analysis of selection, exome-wide dN/dS ratios for somatic substitutions were not significantly different from unity in any species, in line with previous findings in humans[8] (Extended Data Fig. 9). Gene-level analysis did not find genes under significant positive selection in any species, although larger studies are likely to identify rare cancer-driver mutations[8].

## Correlation with life-history traits

Whereas similar mutational processes operate across the species surveyed, the mutation rate per genome per year varied widely. Across the 15 species with age information, we found that substitution rates per genome ranged from 47 substitutions per year in humans to 796 substitutions per year in mice, and indel rates from 2.5 to 158 indels per year, respectively (Fig. 3a, Supplementary Table 4, Methods).

To investigate the relationship between somatic mutation rates, lifespan and other life-history traits, we first estimated the lifespan of each species using survival curves. We used a large collection of mortality data from animals in zoos to minimize the effect of extrinsic mortality (Extended Data Fig. 10). We defined lifespan as the age at which 80% of individuals reaching adulthood have died, to reduce the effects of outliers and variable cohort sizes that affect maximum lifespan estimates[39] (Methods). Notably, we found a tight anticorrelation between somatic mutation rates per year and lifespan across species (Fig. 3b). A log-log allometric regression yielded a strong linear anticorrelation between mutation rate per year and lifespan (fraction of inter-species variance explained (FVE) = 0.85, $P = 1 \times 10^{-6}$), with a slope close to and not significantly different from $-1$. This supports a simple model in which somatic mutation rates per year are inversely proportional to the lifespan of a species (rate $\propto$ 1/lifespan), such that the number of somatic mutations per cell at the end of the lifespan (the end-of-lifespan burden; ELB) is similar in all species.

To further study the relationship between somatic mutation rates and life-history variables, we used linear mixed-effects (LME) regression models. These models account for the hierarchical structure of the data (with multiple crypts per individual and multiple individuals per species), as well as the heteroscedasticity of somatic mutation rate estimates across species (Methods). Using these models, we estimated that the inverse of lifespan explained 82% of the inter-species variance in somatic substitution rates (rate = $k$/lifespan) ($P = 2.9 \times 10^{-9}$; Fig. 3c), with the slope of this regression ($k$) representing the mean estimated ELB across species (3,206.4 substitutions per genome per crypt, 95% confidence interval 2,683.9–3,728.9). Of note, despite uncertainty in the estimates of both somatic mutation rates and lifespans, and despite the diverse life histories of the species surveyed—including around 30-fold variation in lifespan and around 40,000-fold variation in body mass—the estimated mutation load per cell at the end of lifespan varied by only around threefold across species (Table 1). Analogous results were obtained when repeating the analysis with estimates of

the protein-coding mutation rate, which may be a better proxy for the functional effect of somatic mutations (85% of variance explained; ELB: 31 coding substitutions per crypt) (Extended Data Fig. 11, Methods).

We next examined the association between somatic mutation rates and adult body mass, which is known to be a common confounder in correlations that involve lifespan[40,41]. An anticorrelation between somatic mutation rates and body mass may be expected if the modulation of cancer risk across species of vastly different sizes has been a major factor in the evolution of somatic mutation rates. We observed that log-transformed adult body mass was less strongly associated with somatic substitution rates than the inverse of lifespan (allometric regression FVE = 0.21, Fig. 3d; LME regression FVE = 0.44, Fig. 3e). Given that lifespan is correlated with body mass, we performed two tests to assess whether body mass explained any variation in somatic mutation rates that was not explained by lifespan. First, including both the inverse of lifespan and log-transformed adult body mass in the regression model suggested that body mass does not explain a significant amount of variance in somatic mutation rates across species after accounting for the effect of lifespan (likelihood ratio tests: $P = 0.16$ for body mass on a model with lifespan; $P < 10^{-4}$ for lifespan on a model with body mass; Fig. 3f, Methods). Second, partial correlation analyses using allometric regressions further confirmed that the association between somatic mutation rates and lifespan is unlikely to be mediated by the effect of body mass on both variables (lifespan residuals: $P = 3.2 \times 10^{-6}$, FVE = 0.82, Fig. 3b; body mass residuals: $P = 0.39$, FVE = 0.06, Fig. 3d; Methods).

The fact that the variation in somatic mutation rates across species appears to be dominated by lifespan rather than body size is also apparent when looking at particularly informative species. Giraffe and naked mole-rat, for instance, have similar somatic mutation rates (99 and 93 substitutions per year, respectively), in line with their similar lifespans (80th percentiles: 24 and 25 years, respectively), despite a difference of around 23,000-fold in adult body mass (Fig. 3c, e). Similarly, cows, giraffes and horses weigh much more than an average human, and yet have somatic mutation rates that are several fold higher, in line with expectation from their lifespan but not their body mass. Altogether, the weak correlation between body mass and somatic mutation rates after correction for lifespan suggests that the evolution of larger body sizes may have relied on alternative or additional strategies to limit cancer risk, as has been speculated[24,42] (Supplementary Note 2). Of note, the low somatic mutation rate of naked mole-rats, which is unusual for their body mass but in line with their long lifespan (Fig. 3c, e), might contribute to the exceptionally low incidence rates of cancer in this species[43].

We found similar results for other life-history variables that have been proposed to correlate with lifespan, namely basal metabolic rate (BMR) and litter size[44] (Fig. 3f). With the caveat that estimates for these variables vary in quality, they showed weaker correlations with the somatic mutation rate as single predictors, and small non-significant increases in explanatory power when considered together with lifespan (likelihood ratio tests: $P = 0.92$ for litter size; P = 0.083 for log-BMR; $P = 0.79$ for allometric BMR residuals; Fig. 3f, Methods). We note that the results above are robust to the use of alternative measures of the somatic mutation rate, including the rate per exome or mutations per Mb (Extended Data Fig. 11, Methods); alternative estimates of lifespan, including maximum lifespan (Extended Data Fig. 12, Methods); alternative regression models, including a Bayesian hierarchical model and a phylogenetic generalised least-squares regression, which accounts for the effect of phylogenetic relationships (Extended Data Fig. 13a, b, Methods); and bootstrapping analyses at the level of individuals or species (Extended Data Fig. 13c, Methods).

## Mutational processes and lifespan

To investigate whether a single biological process could drive the association between somatic mutation rates and lifespan, we analysed each mutational signature separately. SBS1, SBSB/5 and SBSC/18 are believed to result from different forms of DNA damage and are expected to be

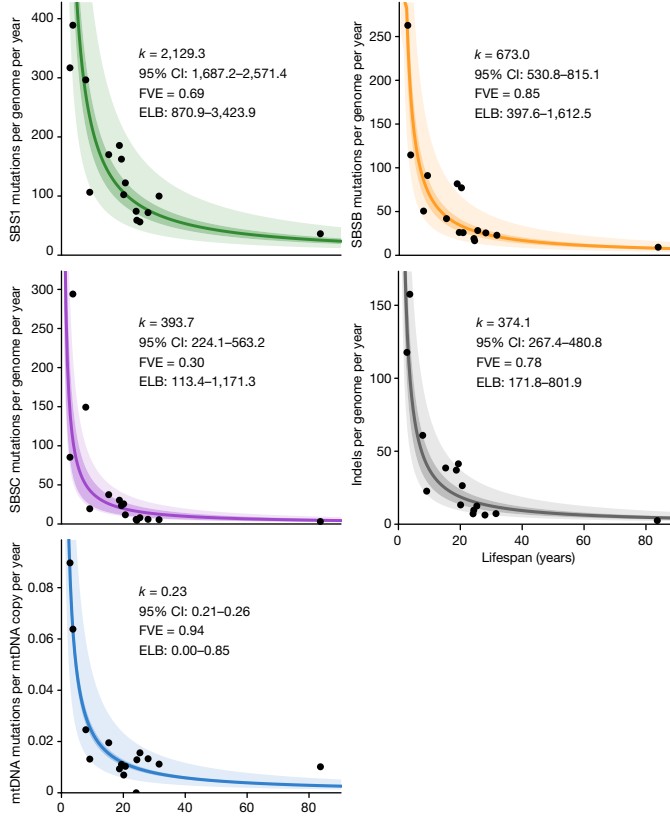

**Fig. 4 | Association between mutation rate subtypes and species lifespan.**
Zero-intercept LME regression of somatic rates of signature-specific
substitutions, indels and mtDNA mutations on inverse lifespan (1/lifespan),
presented on the scale of untransformed lifespan (*x* axis). For simplicity, *y* axes
present mean mutation rates per species, although mutation rates per crypt
were used in the regressions. The darker shaded areas indicate 95% confidence
intervals (CI) of the regression lines, and the lighter shaded areas mark a
twofold deviation from the regression lines. Point estimates and 95% CI of the
regression slope (*k*), fraction of inter-species variance explained by for each
model (FVE) and ranges of end-of-lifespan burden (ELB) are indicated.

subject to different DNA repair pathways[18,33]. They also appear to differ
in their association with the rate of cell division in humans, with SBS1
being more common in fast-proliferating tissues, such as colon and
embryonic or foetal tissues, and SBS5 dominating in post-mitotic cells
in the absence of cell division[14,18,20]. Overall, we found clear anticor-
relations between mutation rates per year and lifespan for the three
substitution signatures and for indels, suggesting that a single biologi-
cal process or DNA repair pathway is unlikely to be responsible for this
association (Fig. 4). The total mutation burden also appears to show a
closer fit with lifespan than individual mutational processes, as meas-
ured by the range of end-of-lifespan burden for each process across
species (Fig. 4). This might be expected if the observed anticorrelation
were the result of evolutionary pressure on somatic mutation rates.

DNA damage and somatic mutations in the mitochondrial genome
have also attracted considerable interest in the ageing field[45]. Our
whole-genome sequencing of individual crypts provided high coverage
of the mitochondrial genome, ranging from 2,188- to 29,691-fold. Nor-
malized against the nuclear coverage, these data suggest that colorectal
crypts contain on the order of around 100–2,000 mitochondrial genomes
per cell (Extended Data Fig. 14a). Using a mutation-calling algorithm that
is sensitive to low-frequency variants, we found a total of 261 mitochon-
drial mutations across 199 crypts (Extended Data Fig. 14a, Methods). The
mutational spectra across species appeared broadly consistent with that
observed in humans, with a dominance of C>T and A>G substitutions

that are believed to result from mitochondrial DNA (mtDNA) replication
errors rather than DNA damage[46] (Extended Data Fig. 14b). Although the
low number of mitochondrial mutations detected per species precludes
a detailed analysis, the estimated number of somatic mutations per copy
of mtDNA also appears to show an anticorrelation with lifespan. Across
species, we obtained an average of 0.23 detectable mutations per copy
of the mitochondrial genome by the end of lifespan (Fig. 4, Methods)—a
considerable burden given the coding-sequence density and the func-
tional relevance of the mitochondrial genome.

## Discussion

Using whole-genome sequencing of 208 colorectal crypts from 56
individuals, we provide insights into the somatic mutational landscape
of 16 mammalian species. Despite their different diets and life histo-
ries, we found considerable similarities in their mutational spectra.
Three main mutational signatures explain the spectra across species,
albeit with varying contributions and subtle variations in the profile
of signature SBSB. These results suggest that, at least in the colorectal
epithelium, a conserved set of mutational processes dominate somatic
mutagenesis across mammals.

The most notable finding of this study is the inverse scaling of somatic
mutation rates with lifespan—a long-standing prediction of the somatic
mutation theory of ageing[11,27]. Considering evolutionary and mechanis-
tic models of ageing together provides a framework for discussing the
possible implications of these results for ageing (see Supplementary
Note 1). Jointly, these models predict ageing to be multifactorial, with
multiple forms of molecular and cellular damage contributing to organ-
ismal ageing owing to evolutionary limits to selection acting on the rates
of these processes. The inverse scaling of somatic mutation rates and
lifespan is consistent with somatic mutations contributing to ageing
and with somatic mutation rates being evolutionarily constrained,
although we discuss alternative explanations below. This interpreta-
tion is also supported by studies reporting more efficient DNA repair
in longer-lived species[47,48]. Somatic mutations could contribute to
ageing in different ways. Traditionally, they have been proposed to
contribute to ageing through deleterious effects on cellular fitness[11,49],
but recent findings question this assumption (Supplementary Note 1).
Instead, the discovery of widespread clonal expansions in ageing human
tissues[19,50–52] raises the possibility that some somatic mutations con-
tribute to ageing by driving clonal expansions of functionally altered
cells at a cost to the organism[49,53,54]. Examples include the possible links
between clonal haematopoiesis and cardiovascular disease[54]; between
mutations in liver disease and insulin resistance[55]; and between driver
mutations in cavernomas and brain haemorrhages[49,53,56]. Detailed stud-
ies on the extent and effect of somatic mutations and clonal expansions
on age-related diseases and ageing phenotypes may help to clarify
the precise role—if any—of somatic mutations in ageing. Even if clear
causal links between somatic mutations and ageing are established,
ageing is likely to be multifactorial. Other forms of molecular damage
involved in ageing could be expected to show similar anticorrelations
with lifespan and, indeed, such anticorrelations have been reported
for telomere shortening and protein turnover[57,58].

Alternative non-causal explanations for the observed anticorrelation
between somatic mutation rates and lifespan need to be considered.
One alternative explanation is that cell division rates could scale with
lifespan and explain the observed somatic mutation rates. Available
estimates of cell division rates, although imperfect and limited to a few
species, do not readily support this argument (Methods). More impor-
tantly, studies in humans have shown that cell division rates are not a
major determinant of somatic mutation rates across human tissues[14,18].
Another alternative explanation for the observed anticorrelation might
be that selection acts to reduce germline mutation rates in species with
longer reproductive spans, which in turn causes an anticorrelation of
somatic mutation rates and lifespan. Although selective pressure on

germline mutation rates could influence somatic mutation rates, it is unlikely that germline mutation rates tightly determine somatic mutation rates: somatic mutation rates in humans are 10–20 times higher than germline mutation rates, show variability across cell types and are influenced by additional mutational processes[18,20]. Overall, the strong scaling of somatic mutation rates with lifespan across mammals, despite the different rates between germline and soma and the variable contributions of different mutational processes across species, suggests that somatic mutation rates themselves have been evolutionarily constrained, possibly through selection on multiple DNA repair pathways. Alternative explanations need to be able to explain the strength of the scaling despite these differences.

Altogether, this study provides a detailed description of somatic mutation across mammals, identifying common and variable features and shedding light on long-standing hypotheses. Scaled across the tree of life and across tissues, in species with markedly different physiologies, life histories, genome compositions and mutagenic exposures, similar studies promise to transform our understanding of somatic mutation and its effects on evolution, ageing and disease.

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

## Methods

### Data reporting

No statistical methods were used to predetermine sample size. The experiments were not randomized and the investigators were not blinded to allocation during experiments and outcome assessment.

### Ethics statement

All animal samples were obtained with the approval of the local ethical review committee (AWERB) at the Wellcome Sanger Institute and those at the holding institutions.

### Sample collection

We obtained colorectal epithelium and skin samples from a range of sources (Supplementary Table 1). For comparability across species an approximately 1-cm biopsy of the colorectal epithelium was taken from the terminal colon during necropsy. All necropsies occurred as soon as possible post-mortem to minimize tissue and DNA degradation. Tissue samples taken later than 24 h post-mortem typically showed extensive degradation of the colorectal epithelium, making the identification of colorectal crypts challenging. These samples were also associated with poor DNA yields and so were not included in the study. Sampled tissue was fixed in PAXgene FIX (PreAnalytiX, Switzerland), a commercially available fixative, during the necropsy. After 24 h in the fixative at room temperature, samples were transferred into the PAXgene STABILIZER and stored at −20 °C until further processing.

### Sample processing

Samples were processed using a workflow designed for detection of somatic mutations in solid tissues by laser-capture microdissection (LCM) using low-input DNA sequencing. For a more detailed description see the paraffin workflow described in another study[29]. In brief, PAXgene-fixed tissue samples of the colorectal epithelium were paraffin-embedded using a Sakura Tissue-Tek VIP tissue processor. Sections of 16 µm were cut using a microtome, mounted on PEN-membrane slides and stained with Gill's haematoxylin and eosin by sequential immersion in the following: xylene (two minutes, twice), ethanol (100%, 1 min, twice), deionized water (1 min, once), Gill's haematoxylin (10 s, once), tap water (20 s, twice), eosin (5 s, once), tap water (20 s, once), ethanol (70%, 20 s, twice) and xylene or Neo-Clear, a xylene substitute (20 s, twice).

High-resolution scans were obtained from representative sections of each species. Example images are shown in Fig. 1a, Extended Data Fig. 2. Individual colorectal crypts were isolated from sections on polyethylene naphthalate (PEN) membrane slides by LCM with a Leica LMD7 microscope. Haematoxylin and eosin histology images were reviewed by a veterinary pathologist. For some samples we also cut a section of muscle tissue from below the colorectal epithelium of the section to use as a germline control for variant calling (Supplementary Table 2). Pre- and post-microdissection images of the tissue were recorded for each crypt and muscle sample taken. Each microdissection was collected in a separate well of a 96-well plate.

Crypts were lysed using the Arcturus PicoPure Kit (Applied Biosystems) as previously described[8,29]. Each crypt then underwent DNA library preparation, without a quantification step to avoid loss of DNA, following a protocol described previously[29]. For some animals, a PAXgene fixed bulk skin biopsy was used as the germline control. For these skin samples, DNA was extracted using the DNeasy Blood & Tissue Kit (Qiagen).

### Library preparation and sequencing

Libraries from microdissected samples were prepared using enzymatic fragmentation, adapter ligation and whole-genome sequencing following a method described previously[29]. Libraries from skin samples were prepared using standard Illumina whole-genome library preparation.

Samples were multiplexed and sequenced using Illumina XTEN and Novaseq 6000 machines to generate 150 base pair (bp) paired-end reads. Samples were sequenced to around 30× depth (Supplementary Table 2).

### Sequence read alignment

For each species, sequences were aligned to a reference assembly (Supplementary Table 2) using the BWA-MEM algorithm[59] as implemented in BWA v.0.7.17-r1188, with options '-T 30 -Y -p -t 8'. The aligned reads were sorted using the bamsort tool from the biobambam2 package (v.2.0.86; gitlab.com/german.tischler/biobambam2), with options 'fixmates=1 level=1 calmdnm=1 calmdnmrecompindetonly=1 calmdnmreference=<reference_fasta> outputthreads=7 sortthreads=7'. Duplicate reads were marked using the bammarkduplicates2 tool from biobambam2, with option 'level = 0'.

### Variant calling

Identification of somatic substitutions and short indels was divided into two steps: variant calling, and variant filtering to remove spurious calls (see 'Variant filtering' below). For human colorectal crypts, we obtained previously sequenced and mapped reads from a study in which colorectal crypts were isolated by LCM[8], and processed them using the sample variant calling and filtering process that was applied to the non-human samples.

Substitutions were identified using the cancer variants through expectation maximization (CaVEMan) algorithm[60] (v.1.13.15). CaVEMan uses a naive Bayesian classifier to perform a comparative analysis of the sequence data from a target and control sample from the same individual to derive a probabilistic estimate for putative somatic substitutions at each site. The copy number options were set to 'major copy number = 5' and 'minor copy number = 2', as in our experience this maximizes the sensitivity to detect substitutions in normal tissues. CaVEMan identifies and excludes germline variants shared in the target (colorectal crypt) and matched normal (skin or muscle tissue) samples, and produces a list of putative somatic mutations that are present only in the target sample. CaVEMan was run separately for each colorectal crypt, using either bulk skin or muscle microdissected from the sample colorectal biopsy as the matched normal control (Supplementary Table 2). For two human donors for whom an alternative tissue was not available, a colonic crypt not included as a target sample was used as the matched normal control.

Indels were identified using the Pindel algorithm[61] (v.3.3.0), using a second sample from the same individual as a matched control. The indel calls produced by Pindel were subsequently re-genotyped using the vafCorrect tool (https://github.com/cancerit/vafCorrect), which performs a local sequence assembly to address alignment errors for indels located at the end of sequence reads, and produces corrected counts of sequence reads supporting the indel and corrected estimates of variant allele fraction (VAF; the fraction of reads supporting the alternate allele at the variant site).

### Variant filtering

A number of post-processing filters were applied to the variant calls to remove false positives (Supplementary Fig. 1a, b).

**Quality flag filter.** CaVEMan and Pindel annotate variant calls using a series of quality flags, with the 'PASS' flag denoting no quality issues affecting the call[60,61]. Variant calls presenting any flag other than 'PASS' were discarded.

**Alignment quality filter.** Variants were excluded if more than half of the supporting reads were clipped. The library preparation methods create short insert size libraries that can result in reads overlapping. To avoid the risk of double counting mutant reads we used fragment-based statistics. Variants without at least four high-quality fragments (alignment

score ≥ 40 and base Phred quality score ≥ 30) were excluded. Variants were excluded if reads supporting the variant had a secondary alignment score that was greater than the primary alignment score. This filter was not applied to indel calls.

**Hairpin filter.** To remove variants introduced by erroneous processing of cruciform DNA during the enzymatic digestion, we applied a custom filter to remove variants in inverted repeats[29]. This filter was not applied to indel calls.

**Chromosome and contig filter.** For species with chromosome-level assemblies, we discarded variants located in non-chromosomal contigs, including the mitochondrial genome (calling of mitochondrial variants is described in the section 'Mitochondrial variant calling and filtering'). For males, variants on the Y chromosome were excluded for species in which the Y chromosome was annotated in the assembly.

**N-tract and contig-end filter.** To reduce artefactual calls due to read misalignment, we discarded variants located within 1 kb of a tract of 50 or more consecutive N bases in the reference assembly, as well as variants within 1 kb of the start or end of a contig (this implies discarding all variants in contigs shorter than 2 kb).

**Sequencing coverage filter.** A sample-specific read depth filter was designed to exclude sites with coverage above the 99th coverage percentile in the sample or its matched normal control, as well as sites with coverage of less than 10× in the sample or its matched normal control.

**Allelic strand bias filter.** We discarded variants without any supporting reads on either the forward or the reverse strand.

**Indel proximity filter.** We discarded variants for which the total number of reads supporting the presence of an indel within 10 bp of the variant was more than three times larger than the number of reads supporting the presence of the variant. This filter was not applied to indel calls.

**Spatial clustering filter.** Visual assessment of variant calls and mutational spectra showed spatially clustered variants to be highly enriched for artefacts. Therefore, we discarded groups of two or more variants located within 1 kb of each other.

**Beta-binomial filter.** For each crypt, an artefact filter based on the beta-binomial distribution was applied, which exploits read count information in other crypts from the same individual. More specifically, for each sample, we fitted a beta-binomial distribution to the variant allele counts and sequencing depths of somatic variants across samples from the same individual. The beta-binomial distribution was used to determine whether read support for a mutation varies across samples from an individual, as expected for genuine somatic mutations but not for artefacts. Artefacts tend to be randomly distributed across samples and can be modelled as drawn from a binomial or a lowly overdispersed beta-binomial distribution. True somatic variants will be present at a high VAF in some samples, but absent in others, and are hence best captured by a highly overdispersed beta-binomial. For each variant site, the maximum likelihood estimate of the overdispersion factor ($\rho$) was calculated using a grid-based method, with values ranging between $10^{-6}$ and $10^{-0.05}$. Variants with $\rho > 0.3$ were considered to be artefactual and discarded. The code for this filter is based on the Shearwater variant caller[62]. We found this to be one of the most effective filters against spurious calls (Supplementary Fig. 1b).

**Minimum VAF filter.** For each sample, we discarded variants for which the VAF was less than half the median VAF of variants passing the beta-binomial filter (see above) in that sample.

**Maximum indel VAF filter.** For each sample, we discarded indels that presented a VAF of greater than 0.9, as such indels were found to be highly enriched in spurious calls in some species. This filter was not applied to substitution calls.

To validate our variant calling strategy, we used LCM to microdissect two sections from the same mouse colorectal crypt. We expected to detect a high fraction of shared somatic variants in these two sections, as their cells should be derived from the same ancestral epithelial stem cell. Both sections were submitted for independent library preparation, genome sequencing, variant calling and filtering using our pipeline. The majority of substitution variant calls (2,742 of 2,933, 93.5%) were shared between both sections (Supplementary Fig. 1c). By contrast, when comparing five separate crypts from a mouse, a maximum of two variants were shared between two crypts, and no variants were shared by three or more crypts (Supplementary Fig. 1d).

### Sample filtering
Our method for estimation of mutation rates assumes monoclonality of colorectal crypt samples. This assumption can be violated owing to several causes, including contamination from other colorectal crypts during microdissection or library preparation, contamination with non-epithelial cells located in or near the crypt, insufficient time for a stem cell to drift to clonality within the crypt, or the possibility that in some species, unlike in humans[8], polyclonal crypts are the norm. Therefore, a truncated binomial mixture model was applied so as to remove crypts that showed evidence of polyclonality, or for which the possibility of polyclonality could not be excluded. An expectation–maximization (EM) algorithm was used to determine the optimal number of VAF clusters within each crypt sample, as well as each cluster's location and relative contribution to the overall VAF distribution. The algorithm considered a range of numbers of clusters (1–5), with the optimal number being that which minimized the Bayesian information criterion (BIC). As the minimum number of supporting reads to call a variant was four, the binomial probability distribution was truncated to incorporate this minimum requirement for the number of successes, and subsequently re-normalized. The EM algorithm returned the inferred optimal number of clusters, the mean VAF (location) and mixing proportion (contribution) of each clone, and an assignment of each input variant to the most likely cluster. After applying this model to the somatic substitutions identified in each sample, sample filtering was performed on the basis of the following three criteria.

**Low mutation burden.** We discarded samples that presented fewer than 50 somatic variants, which was indicative of low DNA quality or sequencing issues.

**High mutation burden.** We discarded samples with a number of somatic variants greater than 3 times the median burden of samples from the same individual (excluding samples with fewer than 50 variants). This served to exclude a small minority of samples that presented evident sequencing quality problems (such as low sequencing coverage), but which did not fulfil the low-VAF criterion for exclusion (see below).

**Low VAF.** We discarded samples in which less than 70% of the somatic variants were assigned to clusters with VAF ≥ 0.3. However, this rule was not applied to those cases in which all the samples from the same individual had primary clusters with mean VAF < 0.3; this was done to prevent the removal of samples from individuals presenting high fractions of non-epithelial cells, but whose crypts were nonetheless dominated by a single clone.

These criteria led to the exclusion of 41 out of 249 samples. On the basis of visual assessment of sequencing coverage and VAF distributions, we decided to preserve three samples (ND0003c_lo0004,

ND0003c_lo0011, TIGRD0001b_lo0010) that we considered to be clonal, but which would have been discarded on the basis of the criteria above.

## Mitochondrial variant calling and filtering

For six species whose reference genome assemblies did not include the mitochondrial sequence, mitochondrial reference sequences were obtained from the GenBank database (Supplementary Table 5). For each species, alignment to the reference genome was performed using BWA (v.0.7.17-r1188), as described above (see 'Sequence read alignment'). Pileup files were generated for mtDNA genomes using the 'bam2R' function in the deepSNV (v.1.32.0) R package[62,63]. The mapping quality cut-off was set to 0, taking advantage of the fact that the mitochondrial genome coverage for most samples was more than 100-fold higher than the nuclear genome coverage, and hence most reads with poor mapping scores should be of mitochondrial origin. Mitochondrial variants were called using the Shearwater algorithm[62] (deepSNV package v.1.32.0). Multiple rounds of filtering were applied to identify and remove false positives. The first set of filters removed germline polymorphisms, applied a maximum false discovery rate (FDR) threshold of $q > 0.01$, required that mismatches should be supported by at least one read on both the forward and reverse strands, and merged consecutive indel calls. Further filtering steps were as follows.

**Minimum VAF filter.** Only variants with VAF > 0.01 were considered for analysis, based on the quality of the mutational spectra.

**Sequencing coverage filter.** Owing to species-specific mtDNA regions of poor mappability, we discarded sites with a read coverage of less than 500×.

**D-loop filter.** Analysis of the distribution of mutations along the mitochondrial genome revealed clusters of mutations within the hypervariable region of mtDNA known as the D-loop. To obtain estimates of the mutation burden in mtDNA unaffected by hypermutation of the D-loop, mutations in the D-loop region (coordinates MT:1–576 and MT:16,024–16,569 in human) were excluded from this analysis.

**High mutation burden.** We discarded samples that had a number of somatic mtDNA variants greater than four times the mean mtDNA burden across all samples. This served to exclude a small minority of samples that were suspected of enrichment in false positive calls. Visual inspection of these samples in a genome browser confirmed the presence of high numbers of variants found on sequence reads with identical start positions and/or multiple base mismatches, suggestive of library preparation or sequencing artefacts.

We examined the mutational spectra of somatic mtDNA substitutions on a trinucleotide sequence context (Extended Data Fig. 14b). The specificity of the filtered variant calls was supported by the observation that the mutational spectra across species were broadly consistent with those previously observed in studies of human tissues[46], with a dominance of C>T and T>C transversions and a strong replication strand bias.

## Mitochondrial copy number analysis

Sequence reads from each sample were separately mapped to the species-specific mtDNA reference sequence to estimate average mtDNA sequencing coverage. Excluding nuclear reference sequences from the alignment enabled even coverage to be obtained across the mitochondrial genome by preventing the mismapping of sequence reads to inherited nuclear insertions of mitochondrial DNA (known as NuMTs). Next, coverage information from individual mtDNA and whole-genome alignment (BAM) files was obtained using the genomecov tool in the bedtools suite (v.2.17.0)[64]. Mitochondrial copy number was calculated according to the formula

$$\text{depth}_{\text{mtDNA}} \times \text{ploidy}/\text{depth}_{\text{gDNA}},$$

where $\text{depth}_{\text{mtDNA}}$ and $\text{depth}_{\text{gDNA}}$ are the mean coverage values for mtDNA and the nuclear genome, respectively, and ploidy = 2 (assuming normal somatic cells to be diploid). For simplicity, the sex chromosomes were excluded from the calculation of the mean nuclear genome coverage.

## Calculation of analysable genome size

To estimate the somatic mutation rate, it was first necessary to establish the size of the analysable nuclear genome (that is, the portion of the genome in which variant calling could be performed reliably) for each sample (Supplementary Table 4). For both substitutions and indels, the analysable genome of a sample was defined as the complement of the union of the following genomic regions: regions reported as 'not analysed' by the CaVEMan variant caller; regions failing the 'chromosome and contig' filter; regions failing the 'N-tract and contig-end' filter; and regions failing the 'sequencing coverage' filter (see 'Variant filtering'). For the analysis of mitochondrial variants, the analysable genome of a sample was defined as the portion of mtDNA that satisfied the 'sequencing coverage' filter (see 'Mitochondrial variant calling and filtering'), after subtracting the hypervariable region (D-loop).

## Life-history data

Obtaining accurate lifespan estimates is challenging; although point estimates of maximum lifespan are available for many species, their veracity is often difficult to assess and estimates can vary widely for the same species (Supplementary Table 6). There can be many causes for this variation, including errors in recording and real variation in longevity between populations (that is, captive versus wild). As we were interested in whether the somatic mutation burden has an association with lifespan in the absence of extrinsic mortality, we sought to obtain estimates of longevity from individuals under human care, to minimize the effect of external factors such as predation or infection.

Mortality records for 14 species were obtained from the Species360 database, authorized by Species360 research data use agreement no. 60633 (Species360 Zoological Information Management System (ZIMS) (2020), https://zims.species360.org). This database contains lifespan data of zoo animals from international zoo records. Using records from 1980 to the present, we excluded animals for which the date of birth or death was unknown or uncertain. To avoid infant mortality influencing the longevity estimates for each species, we removed animals that died before the age of female sexual maturity, as defined by the AnAge database[65]. This resulted in a mean of 2,681 animal lifespan records per species for the species in the study (minimum 309, maximum 8,403; Supplementary Table 6). For the domestic dog, we combined records for domestic dogs (*Canis lupus familiaris*) and wolves (*Canis lupus*), because of the paucity of records for domestic dogs in Species360. Although the data are curated, they are still vulnerable to the presence of inaccurate records, which can bias the lifespan estimates. To reduce the effect of these outliers, for each species lifespan was estimated as the age at which 80% of the adults from that species had died[66] (Supplementary Table 6).

Human longevity estimates were obtained using census birth and death record data from Denmark, (1900–2020), Finland (1900–2019) and France (1900–2018), retrieved from the Human Mortality Database (University of California, Berkeley (USA), and Max Planck Institute for Demographic Research (Germany); https://www.mortality.org, https://www.humanmortality.de). We selected these countries because they had census records going back at least 100 years. To remove the effect of infant mortality, we excluded individuals who died before the age of 13. For each country, we selected the cohort born in 1900 and calculated the age at which 80% of the individuals had died (Denmark, 87 years; Finland, 83 years; France, 81 years). We then used the mean

of the three countries as our estimate of the human 80% lifespan (83.7 years) (Supplementary Table 6).

To test the effects of different estimates of lifespan on our results, we also obtained maximum longevity estimates for each species from a range of databases[67] and a survey of the literature (Supplementary Table 6). Other life-history metrics were obtained from the AnAge database[65] (Supplementary Table 6).

## Mutational signature analysis

Mutational signatures of substitutions on a trinucleotide sequence context were inferred from sets of somatic mutation counts using the sigfit (v.2.1.0) R package[31]. Initially, signature extraction was performed de novo for a range of numbers of signatures ($N$ = 2,...,10), using counts of mutations grouped per sample, per individual and per species. To account for differences in sequence composition across samples, and especially across species, mutational opportunities per sample, per individual and per species were calculated from the reference trinucleotide frequencies across the analysable genome of each sample (see 'Calculation of analysable genome size'), and supplied to the 'extract_signatures' function in sigfit. The 'convert_signatures' function in sigfit was subsequently used to transform the extracted signatures to a human-relative representation (Fig. 2b), by scaling the mutation probability values using the corresponding human genome trinucleotide frequencies. The best-supported number of signatures, on the basis of overall goodness-of-fit[31] and consistency with known COSMIC signatures (https://cancer.sanger.ac.uk/signatures/), was found to be $N$ = 3. The cleanest deconvolution of the three signatures was achieved when using the mutation counts grouped by species, rather than by sample or individual. The three extracted signatures (labelled SBSA, SBSB and SBSC) were found to be highly similar to COSMIC signatures SBS1 (cosine similarity 0.96), SBS5 (0.89) and SBS18 (0.91), respectively. These signatures were independently validated using the Mutational-Patterns (v.1.12.0) R package[68], which produced comparable signatures (respective cosine similarities 0.999, 0.98 and 0.89).

This de novo signature extraction approach, however, failed to deconvolute signatures SBSA and SBSB entirely from each other, resulting in a general overestimation of the exposure to SBSA (Extended Data Fig. 15). To obtain more accurate estimates of signature exposure, the deconvolution was repeated using an alternative approach that combines signature fitting and extraction in a single inference process[31]. More specifically, the 'fit_extract_signatures' function in sigfit was used to fit COSMIC signature SBS1 (retrieved from the COSMIC v.3.0 signature catalogue; https://cancer.sanger.ac.uk/signatures/) to the mutation counts grouped by species (with species-specific mutational opportunities), while simultaneously extracting two additional signatures de novo (SBSB and SBSC). Before this operation, COSMIC SBS1 was transformed from its human-relative representation to a genome-independent representation using the 'convert_signatures' function in sigfit. By completely deconvoluting SBS1 and SBSB, this approach yielded a version of SBSB that was more similar to COSMIC SBS5 (cosine similarity 0.93); the similarity of SBSC to COSMIC SBS18 was the same under both approaches (0.91).

Finally, the inferred signatures were re-fitted to the mutational spectra of mutations in each sample (using the 'fit_signatures' function in sigfit with sample-specific mutational opportunities) to estimate the exposure of each sample to each signature. The fitting of the three signatures yielded spectrum reconstruction similarity values (measured as the cosine similarity between the observed mutational spectrum and a spectrum reconstructed from the inferred signatures and exposures) with median 0.98 and interquartile range 0.96–0.99. Although the purely de novo extraction approach and the 'fitting and extraction' approach yielded comparable versions of signatures SBSB and SBSC, the fixing of COSMIC SBS1 in the latter approach resulted in lower SBS1 exposures and higher SBSB exposures in most samples, owing to the cleaner deconvolution of these two signatures (Fig. 2, Extended Data Fig. 15).

To examine potential variation in the spectrum of signature SBS5 across species, the following procedure was conducted for each species: individual-specific mutation counts and mutational opportunities were calculated for each individual in the species, and the 'fit_extract_signatures' function was used to fit COSMIC signatures SBS1, SBS18 and SBS34 (transformed to a genome-independent representation using the 'convert_signatures' function) to the mutational spectra of each individual, while simultaneously inferring one additional signature (corresponding to signature SBS5 as manifested in that species; Extended Data Fig. 6).

To assess the presence in non-human colorectal crypts of mutational signatures caused by APOBEC or colibactin, which have been previously observed in human crypts[8], we used an expectation–maximization algorithm for signature fitting, in combination with likelihood ratio tests (LRTs). More specifically, for each non-human sample, we tested for exposure to colibactin (signature SBS88, COSMIC v.3.2) by comparing the log-likelihoods of (i) a model fitting COSMIC signatures SBS1, SBS5, SBS18, SBS34 and SBS88, and (ii) a reduced model fitting only the first four signatures. Benjamini–Hochberg multiple-testing correction was applied to the $P$ values that resulted from the LRTs, and colibactin exposure was considered significant in a sample if the corresponding corrected $q$-value was less than 0.05. We followed the same approach to assess exposure to APOBEC (SBS2 and SBS13), using two separate sets of LRTs for models including either SBS2 or SBS13, in addition to SBS1, SBS5, SBS18 and SBS34. APOBEC exposure was considered significant in a sample if its $q$-values for the models including SBS2 and SBS13 were both less than 0.05. This analysis identified 1/180 crypts with significant exposure to each of colibactin and APOBEC (although the evidence for the presence of the relevant signatures in these two crypts was not conclusive). To test for depletion of colibactin or APOBEC exposure in non-human crypts relative to human crypts, we first applied the LRT-based method described above to a published set of 445 human colorectal crypts[8], identifying 92 colibactin-positive and 9 APOBEC-positive crypts. We then compared the numbers of colibactin- and APOBEC-positive crypts in the human and non-human sets using two separate Fisher's exact tests ('fisher.test' function in R). This revealed the difference in colibactin exposure to be highly significant ($P = 7 \times 10^{-14}$), unlike the difference in APOBEC exposure ($P = 0.30$).

Mutational spectra of somatic indels identified in each species were generated using the 'indel.spectrum' function in the Indelwald tool for R (24/09/2021 version; https://github.com/MaximilianStammnitz/Indelwald).

## Selection analysis

Evidence of selection was assessed using the ratio of nonsynonymous to synonymous substitution rates (dN/dS) in the somatic mutations called in each species. The dNdScv (v.0.0.1.0) R package[38] was used to estimate dN/dS ratios for missense and truncating substitutions in each species separately. Reference CDS databases for the dNdScv package were built for those species with available genome annotation in Ensembl (https://www.ensembl.org; Supplementary Table 2), using the 'buildref' function in dNdScv. For each species, the 'dndscv' function was applied to the list of somatic substitutions called in samples of that species, after de-duplicating any substitutions that were shared between samples from the same individual to avoid counting shared somatic mutations multiple times. In addition, the analysis was restricted to genes that were fully contained in the analysable genomes of all samples from the species (a condition satisfied by the vast majority of protein-coding genes). Genome-wide and gene-specific dN/dS ratios were obtained for missense and truncating substitutions in each species; no genes with statistically significant dN/dS ≠ 1 were observed.

## Copy number analysis

For species with chromosome-level assemblies (cat, cow, dog, horse, human, mouse, rabbit and rat), the total and the allele-specific copy

number (CN) was assessed in each sample, adapting a likelihood model that was previously applied to the detection of subclonal CN changes in healthy human skin[19]. This method exploits two sources of evidence: relative sequencing coverage and B-allele fraction (BAF; the fraction of reads covering a heterozygous single-nucleotide polymorphism (SNP) that support one of the alleles). Human samples PD36813x15 and PD36813x16 were excluded from this analysis owing to the poor quality of their SNP data.

For each sample, sequencing coverage was measured in non-overlapping 100-kb bins along the reference genome of the species, using the coverageBed tool in the bedtools suite (v.2.17.0)[64]. For each bin, the coverage per base pair was calculated by dividing the number of reads mapping to the bin by the bin length, and multiplying the result by the read length (150 bp). A normalized sample–normal coverage ratio was then calculated for each bin by dividing the bin coverage in the sample by the corresponding coverage in its matched normal control (see 'Sample processing'). Heterozygous SNPs were isolated for each sample by selecting germline SNPs with a BAF between 0.4 and 0.6 in the matched normal sample, and a coverage of at least 15 reads in both the target sample and its matched normal sample. After assigning each SNP to its corresponding 100-kb genome bin, the bins in each sample were divided into two sets: (i) bins with coverage ≥ 10 in both the target sample and its matched normal, and at least one heterozygous SNP; and (ii) bins with coverage ≥ 10 in both the target sample and its matched normal, and no heterozygous SNPs. For the first set, estimates of total and allele-specific CN were inferred by maximizing the joint likelihood of a beta-binomial model for BAF and a negative binomial model for relative coverage, as previously described[19]. The most likely combination of allele CN values was obtained for each bin by conducting an exhaustive search of CN values between 0 and 4, and selecting the combination maximizing the joint likelihood (calculated on the basis of expected BAF and relative coverage values). A penalty matrix was used to penalize more complex solutions over simpler ones, as previously described[19]. For the second set of bins (bins without SNPs), only estimates of total CN were inferred, by maximizing the likelihood of a negative binomial model for relative coverage. The most substantial differences between these methods and the one previously published are: (i) SNPs were obtained from the variant calling output, instead of from a public database; (ii) relative coverage was calculated per 100-kb bin, rather than per SNP; (iii) SNPs were not phased within each gene, but within each bin; (iv) no reference bias was assumed (that is, the underlying BAF of heterozygous SNPs was assumed to be 0.5); (v) the minimum sample purity was raised to 0.85; (vi) putative CN changes were not subjected to significance testing, but selected according to their likelihood, and subsequently filtered by means of a segmentation algorithm (see below).

Estimates of total and allele-specific CN per bin were merged into CN segments, which were defined as contiguous segments composed of five or more bins with identical CN states. Segmentation was performed separately for total and allele-specific CN estimates in each sample. After this process, any pair of adjacent segments with the same CN assignment, and separated by a distance shorter than five bins, was merged into a single segment. Finally, within each species, segments presenting CN values other than 2 (or 1/1 for allele-specific CN), and being either shorter than 10 bins (1 Mb), or shared among two or more samples, were discarded, resulting in the removal of nearly all spurious CN changes.

## Estimation of mutation rate

For each sample, the somatic mutation density (mutations per bp) was calculated by dividing the somatic mutation burden (total number of mutations called) by the analysable genome size for the sample (see 'Calculation of analysable genome size'). The adjusted somatic mutation burden (number of mutations per whole genome) was then calculated by multiplying the mutation density by the total genome size of the species (see below). The somatic mutation rate per year

(mutations per genome per year) was obtained by dividing this adjusted mutation burden by the age of the individual, expressed in years (Supplementary Table 2). The expected ELB for each sample was calculated by multiplying the somatic mutation rate by the estimated lifespan of the species (see 'Life-history data').

The total genome size of a species was estimated as the total size of its reference genome assembly. Across species, the mean genome size was 2.67 Gb, ranging between 2.41 Gb and 3.15 Gb and with a standard deviation of 221 Mb (Supplementary Table 4). This suggests that inter-species variation in genome size should not have a substantial influence on the somatic mutation rate estimates. For an assessment of alternative measures of mutation rate, see 'Association of mutation rate and end-of-lifespan burden with lifespan'.

## Association of mutation rate with life-history traits

The association of the somatic mutation rate with different life-history traits was assessed using LME models. In particular, associations with the following traits were examined: lifespan (in years), adult mass (or adult weight, in grams), BMR (in watts), and litter size (see 'Life-history data'). Associations for lifespan, adult mass and BMR were assessed using the following transformed variables: 1/lifespan, $\log_{10}$(adult mass) and $\log_{10}$(BMR). To account for the potentially confounding effect of the correlation between metabolic rate and body mass, the residuals of a fitted allometric regression model of BMR on adult mass (equivalent to a simple linear regression of $\log_{10}$(BMR) on $\log_{10}$(adult mass)) were used as a mass-adjusted measure of metabolic rate, referred to as 'BMR residuals'.

For each variable, an LME model was implemented for the regression of somatic mutation rates per sample on the variable of interest, using the 'lme' function in the nlme R package (v.3.1-137; https://cran.r-project.org/web/packages/nlme). To account for non-independence of the samples, both at the individual level and at the species level, the model included fixed effects (intercept and slope parameters) for the variable of interest, and random effects (slope parameters) at the individual and species levels. In addition, to account for the heteroscedasticity of mutation rate estimates across species, the usual assumption of constant response variance was replaced with explicit species-specific variances, to be estimated within the model.

To determine the fraction of inter-species variance in mutation rate explained by each life-history variable individually, the LME model described above was used to produce predictions of the mean mutation rate per species; only fixed effects were used when obtaining these predictions, random effects being ignored. The variance of these predictions was then compared to the variance in observed mean mutation rates; the latter were calculated for each species as the mean of the observed mean rates per individual, to avoid individuals with larger numbers of samples exerting a stronger influence on the species mean. The fraction of inter-species variance explained by the model was calculated using the standard formula for the coefficient of determination,

$$R^2 = \text{ESS}/(\text{ESS} + \text{RSS}),$$

where ESS is the explained sum of squares, and RSS is the residual sum of squares:

$$\text{ESS} = \sum_i (\hat{y}_i - \bar{y})^2, \text{RSS} = \sum_i (y_i - \hat{y}_i)^2.$$

In this formulation, $y_i$ and $\hat{y}_i$ denote the observed and predicted mutation rates for species $i$, respectively, and $\bar{y}$ is the overall mean rate. This definition of $R^2$ coincides with the fraction of variance explained (FVE), defined as 1 minus the fraction of variance unexplained (FVU):

$$\text{FVE} = 1 - \text{FVU} = 1 - [\text{RSS}/(\text{ESS} + \text{RSS})] = \text{ESS}/(\text{ESS} + \text{RSS}) = R^2.$$

As the predicted and observed values correspond to mean mutation rates per species, rather than mutation rates per sample, FVE provides a

measure of the fraction of inter-species variance explained by the fixed effects of the LME model. Among the variables considered, 1/lifespan was found to have the greatest explanatory power (FVE = 0.84, using a free-intercept model).

To compare the explanatory power of variables other than 1/lifespan when considered either individually or in combination with 1/lifespan, the method described above was also applied to two-variable combinations of 1/lifespan and each of the remaining variables, using an LME model with fixed effects for both variables and random effects for 1/lifespan only. The $R^2$ formula above was used to measure the fraction of inter-species variance explained by each model. In addition, to test whether the inclusion of a second explanatory variable was justified by the increase in model fit, LRTs between each two-variable LME model and a reduced LME model including only 1/lifespan were performed using the 'anova' function in the nlme R package.

To further assess the potential effects of body mass and lifespan on each other's association with the somatic mutation rate, allometric regression models (equivalent to simple linear models under logarithmic transformation of both variables) were fitted to the mean somatic mutation rate per species, using either adult mass or lifespan as the explanatory variable. In addition, the 'allometric residuals' of mutation rate, adult mass and lifespan (that is, the residuals of pairwise allometric regressions among these three variables) were used to examine the associations between somatic mutation rate and either body mass or lifespan, after accounting for the effect of the third variable (partial correlation analysis). For example, to account for the potential influence of body mass on the relationship between somatic mutation rate and lifespan, the residuals of an allometric regression between mutation rate and adult mass, and the residuals of an allometric regression between lifespan and adult mass, were analysed using simple linear regression. This analysis supported a strong association between somatic mutation rate and lifespan (independently of the effect of mass; FVE = 0.82, $P = 3.2 \times 10^{-6}$; Fig. 3c), and a non-significant association between somatic mutation rate and body mass (independently of the effect of lifespan). Therefore, the relationship between somatic mutation rate and lifespan does not appear to be mediated by the effect of body mass on both variables. Of note, this result remains after excluding naked mole-rat: after removing this species, partial correlation analysis still reveals a strong association between somatic mutation rate and lifespan (FVE = 0.77, $P = 4.1 \times 10^{-5}$), and a non-significant association between somatic mutation rate and body mass ($P = 0.84$). This demonstrates that the observed relationships are not dependent on the presence of naked mole-rat in the study.

To assess the robustness of the LME regression analyses described above, we performed bootstrap analysis on each LME model, at the level of both individuals and species. More specifically, for each level we used each of the LME models to perform regression on 10,000 bootstrap replicates, produced by resampling either species or individuals with replacement. We then assessed the distributions of FVE across bootstrap replicates (Extended Data Fig. 13c). In addition, we performed a similar bootstrap analysis using a collection of maximum longevity estimates obtained from the literature (see 'Life-history data'). We applied the zero-intercept LME model described above (for regressing mutation rate on inverse lifespan) on a set of 5,000 bootstrap replicates, each of which used a set of species lifespan estimates randomly sampled from the collection of literature-derived estimates (Extended Data Fig. 12).

The results obtained with the LME models were additionally validated using an independent hierarchical Bayesian model, in which the mean somatic mutation burden of each individual was modelled as following a normal distribution with mean defined as a linear predictor containing a species-specific slope parameter and a multiplicative offset (corresponding to the individual's age; inclusion of this offset minimizes the heteroscedasticity of the mutation rate across species, which results from dividing mutation burdens by age). Species-specific slope parameters were in turn modelled as normally distributed around a global slope

parameter, equivalent to the fixed-effect slope estimated by the LME model. This hierarchical model produced very similar results to those of the LME model for all life-history variables (Extended Data Fig. 13a).

We note that samples CATD0002b_lo0003 and MD6267ab_lo0003 were excluded from all regression analyses, owing to the fact that each shared the most of its somatic variants with another sample from the same individual (indicating, in each case, that both samples were closely related), hence violating the assumption of independence among samples. The inclusion of these two samples, however, had no effect on the outcome of the analyses.

## Association of mutation rate and end-of-lifespan burden with lifespan

The relationship between somatic mutation rate and species lifespan was further explored by adapting the LME model described in the previous section to perform constrained (zero-intercept) regression of the adjusted mutation rate per year on the inverse of lifespan, 1/lifespan (see 'Life-history data', 'Estimation of mutation rate' and 'Association of mutation rate with life-history traits'). The use of zero-intercept regression was motivated by the prediction that, if somatic mutation is a determinant of maximum lifespan, then it would be expected for all species to end their lifespans with a similar somatic mutation burden. Indeed, this was confirmed by simple linear regression of the species mean end-of-lifespan mutation burden against species lifespan (slope $P = 0.39$). Thus, if $m$ is the mutation rate per year, and $L$ is the species' lifespan, the expected relationship is of the form.

$$m \, L \approx k,$$

where $k$ is a constant representing the typical end-of-lifespan mutation burden across species. According to this relationship, the mutation rate per year is linearly related to the inverse of lifespan,

$$m \approx k(1/L).$$

Therefore, the cross-species average end-of-lifespan burden ($k$), can be estimated as the slope parameter of a zero-intercept linear regression model with the mutation rate per year ($m$) as the dependent variable, and the inverse of lifespan ($1/L$) as the explanatory variable. To this purpose, the LME model described in the previous section was altered by removing the fixed-effect intercept parameter, thus considering only fixed- and random-effect slope parameters for 1/Lifespan.

The zero-intercept LME model estimated a value of $k = 3,210.52$ (95% confidence interval 2,686.89–3,734.15). The fraction of inter-species variance explained by the zero-intercept model (FVE) was 0.82, whereas the LME model described in the previous section (which estimated $k = 2,869.98$, and an intercept of 14.76) achieved FVE = 0.84 (see 'Association of mutation rate with life-history traits'). To test whether the increase in model fit justifies the inclusion of an intercept, both models were compared using a LRT (as implemented by the 'anova' function in the nlme R package (v.3.1-137)). This yielded $P = 0.23$, indicating that the free-intercept model does not achieve a significantly better fit than the zero-intercept model. Similarly, the zero-intercept model yielded lower values for both the Bayesian information criterion (BIC) and the Akaike information criterion (AIC). Notably, equivalent analyses using somatic mutation rates per megabase and per protein-coding exome (instead of per whole genome) yielded comparable results (Extended Data Fig. 11).

To investigate the possibility of phylogenetic relationships between the species sampled confounding the analysis, a phylogenetic generalized linear model was used to regress the mean mutation rate of each species on the inverse of its lifespan ($1/L$), while accounting for the phylogenetic relationships among species. A phylogenetic tree of the 15 species examined was obtained from the TimeTree resource[69], and the phylogenetic linear model was fitted using the 'pgls' function

in the caper R package (v.1.0.1; https://cran.r-project.org/web/pack-ages/caper). The estimates produced by zero-intercept regression of mean mutation rates per species on 1/lifespan were compared between this phylogenetic generalized linear model and a simple linear model ('lm' function in R). The use of this simple model, as well as the use of mean mutation rates per species (rather than mutation rates per sample), was necessary owing to the impossibility of replicating the heteroscedastic mixed-effects structure of the LME model used for the main association analyses (see 'Association of mutation rate with life-history traits') within the phylogenetic linear model. Both the phylogenetic linear model and the simple linear model produced similar estimates (Extended Data Fig. 13b), suggesting that phylogenetic non-independence of the samples does not have a substantial effect on the association analyses.

### Cell division analysis

To investigate the extent to which differences in cell division rate could explain differences in mutation rate and burden across species, we obtained estimates of intestinal crypt cell division rates from mouse[70], rat[71] and human[72,73] (Supplementary Table 7). Using these cell division rates, our lifespan estimates and the observed substitution rates, we calculated the number of cell divisions at the end of lifespan and the corresponding number of mutations per cell division expected under a simple model assuming that all mutations occur during cell division (Supplementary Table 7).

We investigated whether differences in the number of cell divisions among species could explain the observed differences in mutation burden. Although colorectal cell division rate estimates are lacking for most species, existing estimates from mouse, rat and human indicate that the total number of stem cell divisions per crypt in a lifetime varies greatly across species—for example, there are around 6- to 31-fold more divisions per intestinal stem cell in a human than in a rat over their respective lifetimes, depending on the estimate of cell division rate used (Supplementary Table 7). Mouse intestinal stem cells are estimated to divide once every 24 h (ref. [70]), whereas estimates of the human intestinal stem cell division rate vary from once every 48 h (ref. [72]) to once every 264 h (ref. [73]). Thus, mouse intestinal stem cells divide 2–11 times faster than human intestinal stem cells. By the end of lifespan, an intestinal stem cell is predicted to have divided around 1,351 times in a mouse, around 486 times in a rat and 2,774–15,257 times in a human (depending on the estimate of cell division rate used). Applying our somatic mutation burden and lifespan data, this implies that the somatic mutation rate per cell division in a mouse is around 1.5- to 8.4-fold higher than in a human. However, the observed fold difference in somatic mutation rate between these two species is 16.9 (Table 1). Therefore, differences in cell division rate appear unable to fully account for the observed differences in mutation rate across species. Nevertheless, we note that accurate cell division rate estimates for basal intestinal stem cells are lacking for most species.

### Reporting summary

Further information on research design is available in the Nature Research Reporting Summary linked to this paper.

### Data availability

DNA sequence data have been deposited in the European Genome-Phenome Archive (EGA; https://ega-archive.org) under overarching accession EGAD00001008032. Human DNA sequence data from a previous study[8] are deposited in the EGA (accession EGAD00001004192). Processed mutation calls and other data used in the analyses have been deposited in Zenodo (https://doi.org/10.5281/zenodo.5554777). Raw mortality data used to estimate lifespan (Species360 Data Use Approval Number 60633) cannot be publicly shared, as Species360 is the custodian (not the owner) of their members' data. Raw data are accessible through Research Request applications to Species360. Once Species360 grants access to data, they are intended only for and restricted to use in the project they were approved for and for a single publication. Any email communications should be directed to support@species360.org.

### Code availability

The computer code used in the analyses has been deposited in Zenodo (https://doi.org/10.5281/zenodo.5554801) and GitHub (https://github.com/baezortega/CrossSpecies2021).

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

**Acknowledgements** We thank the staff of the Wellcome Sanger Institute for help in this project, including the Scientific Operations, Informatics and CASM support teams. This research was made possible by the worldwide information network of zoos and aquariums that are members of Species360, and is authorized by Species360 research data use and grant agreement 60633. We thank D. Conde and J. Staerk for help accessing Species360 data; and J. P. de Magalhaes, J. DeGregori, J. Vijg and M. Lynch for their comments on the manuscript. This research was funded by Wellcome (grant number 206194), the Dunhill Medical Trust (RPGF2002\188) and the Deutsche José Carreras Leukämie-Stiftung. I.M. is funded by Cancer Research UK (C57387/A21777) and the Wellcome Trust. P.J.C. is a Wellcome Trust Senior Clinical Fellow.

**Author contributions** A.C., E.P.M., M.R.S. and I.M. conceived the project. I.M., E.P.M. and M.R.S. supervised the project. E.F., S.S., I.J., E.W., N.M., R.D., M.W.P., M.D., R.B., M.D.M., F.G., F.C.-C., L.P., D.B., E.St.J.S., B.N. and E.P.M. performed and facilitated sample collection. A.C. performed the laser capture microdissection. A.C., L.M.R.H., A.R.J.L., Y.H., K.R., E.A. and S.L. processed the samples. A.C., A.B.-O., F.A., N.B., T.H.H.C., M.A.S., D.J., R.E.A. and S.B. processed the data. A.C., A.B.-O. and N.B. led the analysis with help from F.A., T.H.H.C, M.A.S., A.R.J.L., T.M.B., T.D., H.J., E.P.M. and I.M. The manuscript was written by A.C., A.B.-O., N.B. and I.M., with input from all of the authors.

**Competing interests** The authors declare no competing interests.

**Additional information**
**Correspondence and requests for materials** should be addressed to Alex Cagan or Iñigo Martincorena.

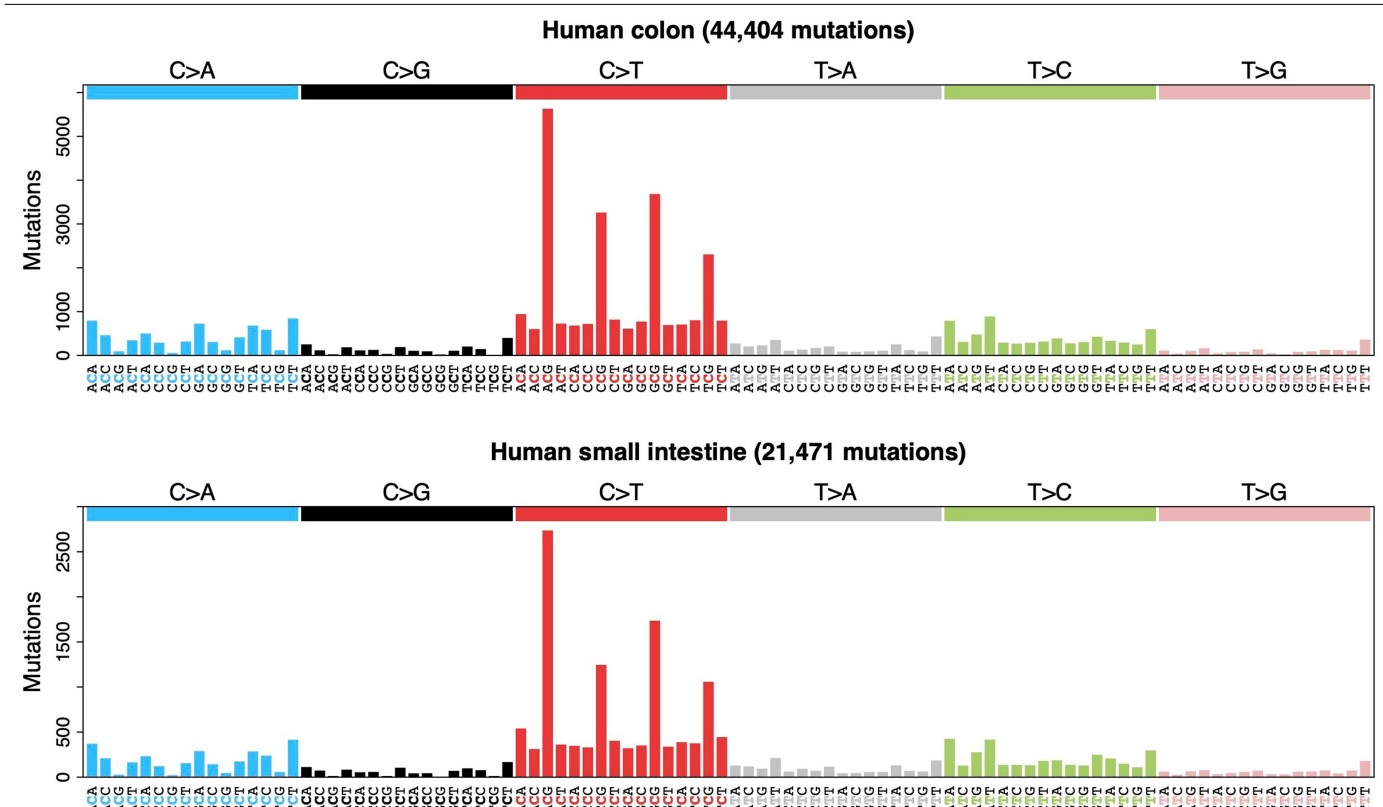

**Extended Data Fig. 1 | Somatic mutational spectra of human colon and small intestine.** Trinucleotide-context mutational spectra of somatic substitutions from human adult stem cells in colon (top) and small intestine, using mutation calls obtained from a previous study[14].

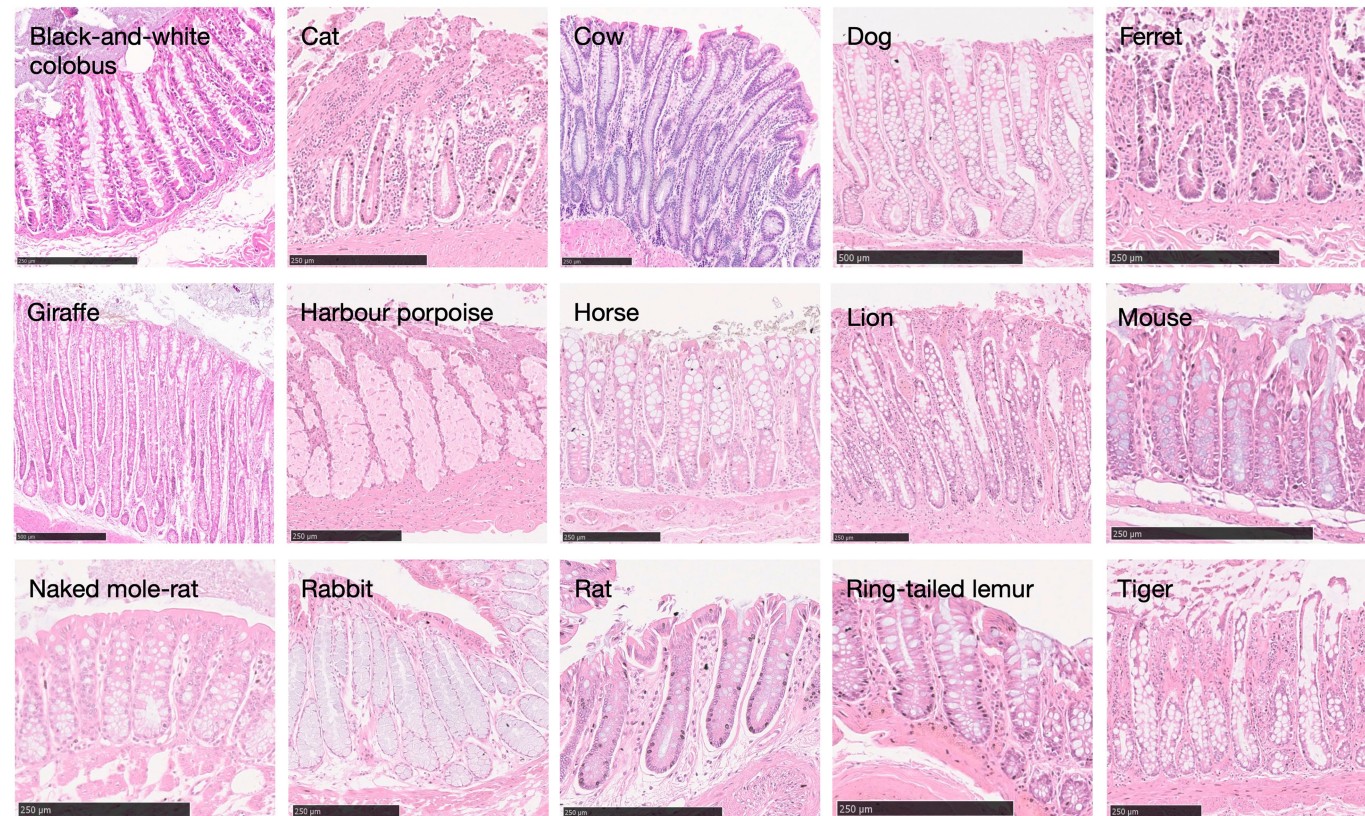

**Extended Data Fig. 2 | Histology images of intestinal crypts across species.** Histological images of the colorectal or intestinal (ferret) epithelium for each non-human species. Scale bars are provided at the bottom of each image.

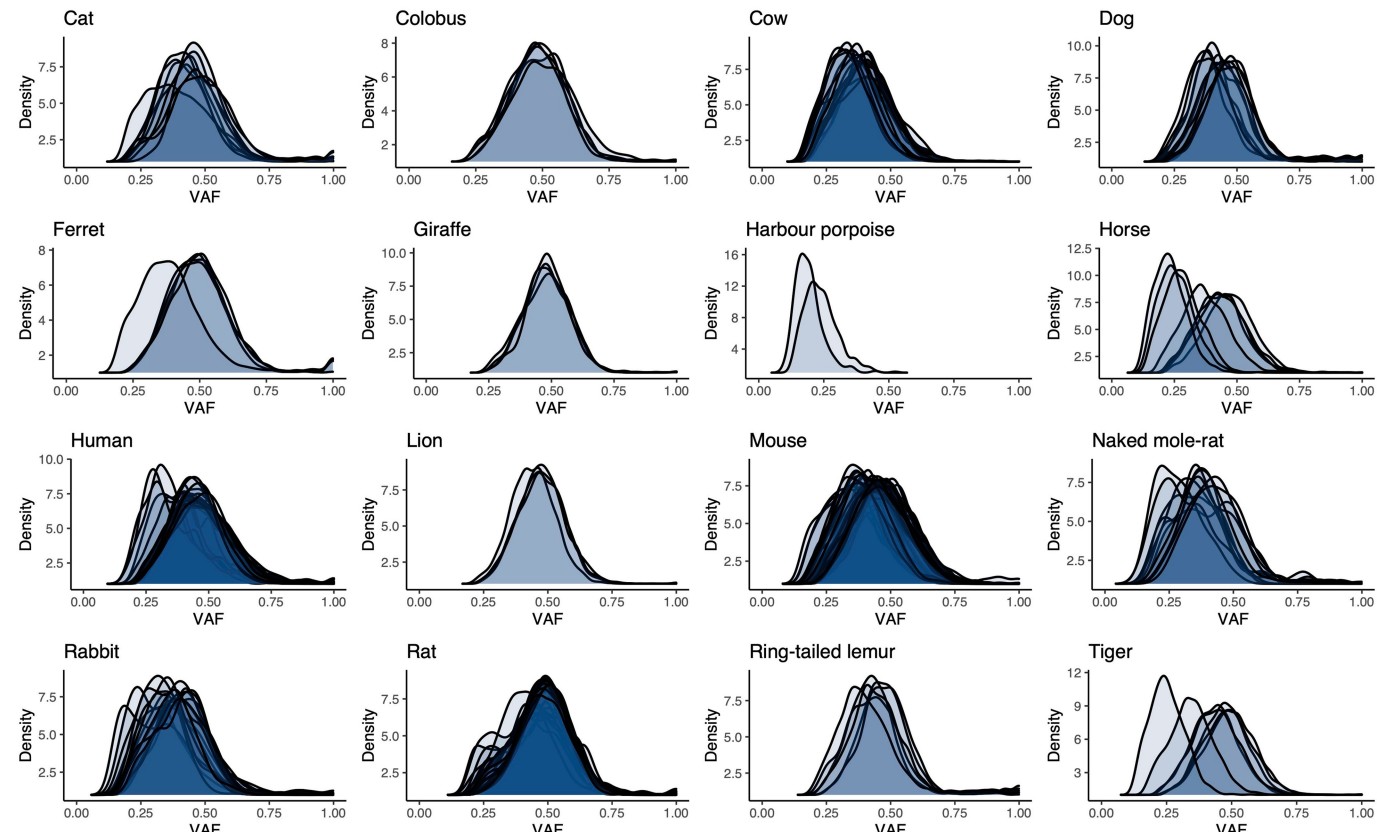

**Extended Data Fig. 3 | Somatic VAF distributions per species.** Distributions of variant allele fraction (VAF) for somatic substitutions in each crypt for each species. Each distribution refers to the variants in a single sequenced crypt.

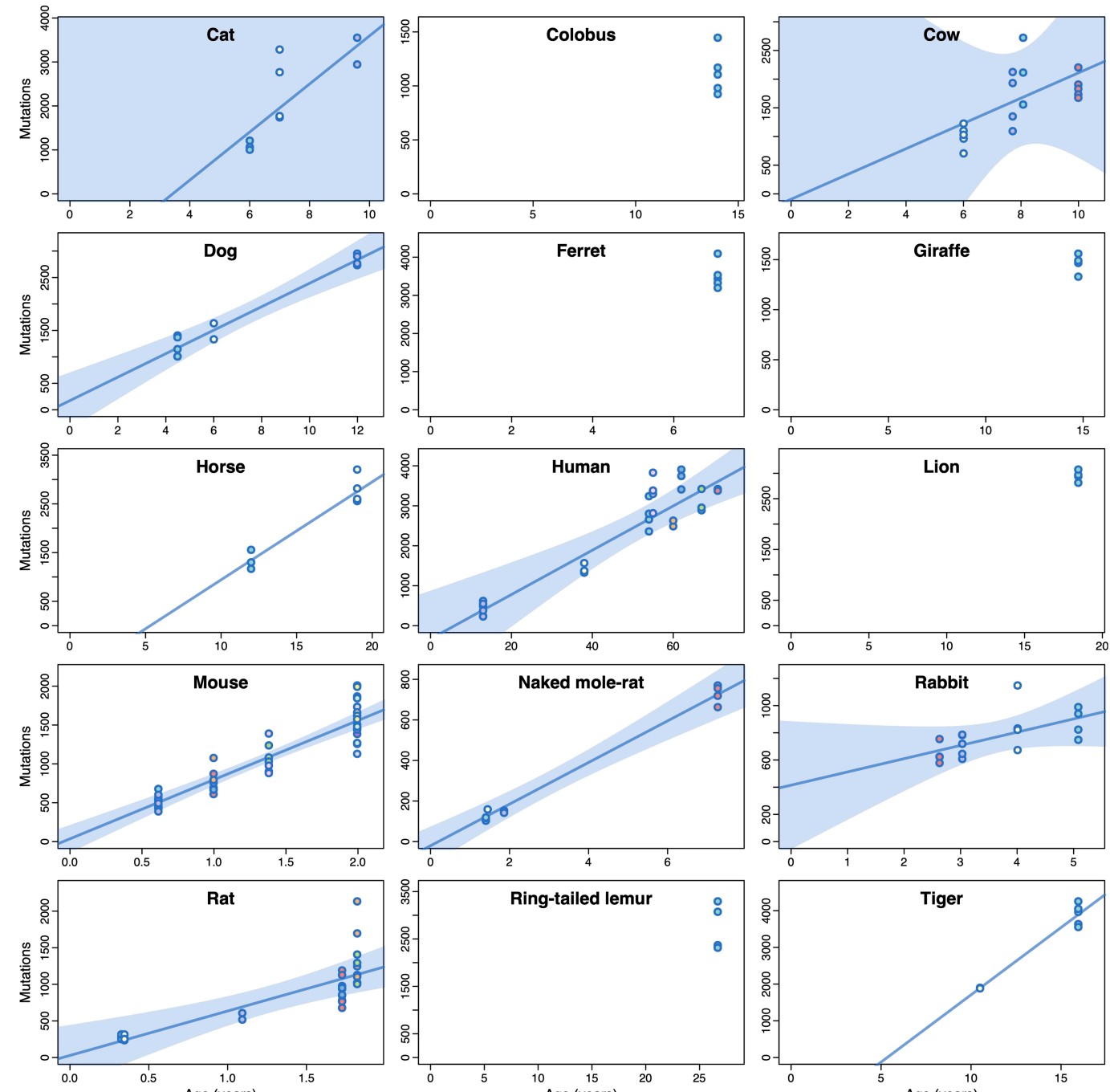

**Extended Data Fig. 4 | Somatic mutation accumulation across species.** Each panel presents somatic substitution burdens per genome (corrected for analysable genome size) for a given species. Each dot represents a crypt sample, with samples from the same individual sharing the same colour. For species with two or more individuals, the estimated regression line from a simple linear regression model on individual mean burdens is shown. For species with three or more individuals, the shaded region indicates 95% confidence intervals of the regression line. Harbour porpoise samples were excluded owing to unknown age of the sampled individual.

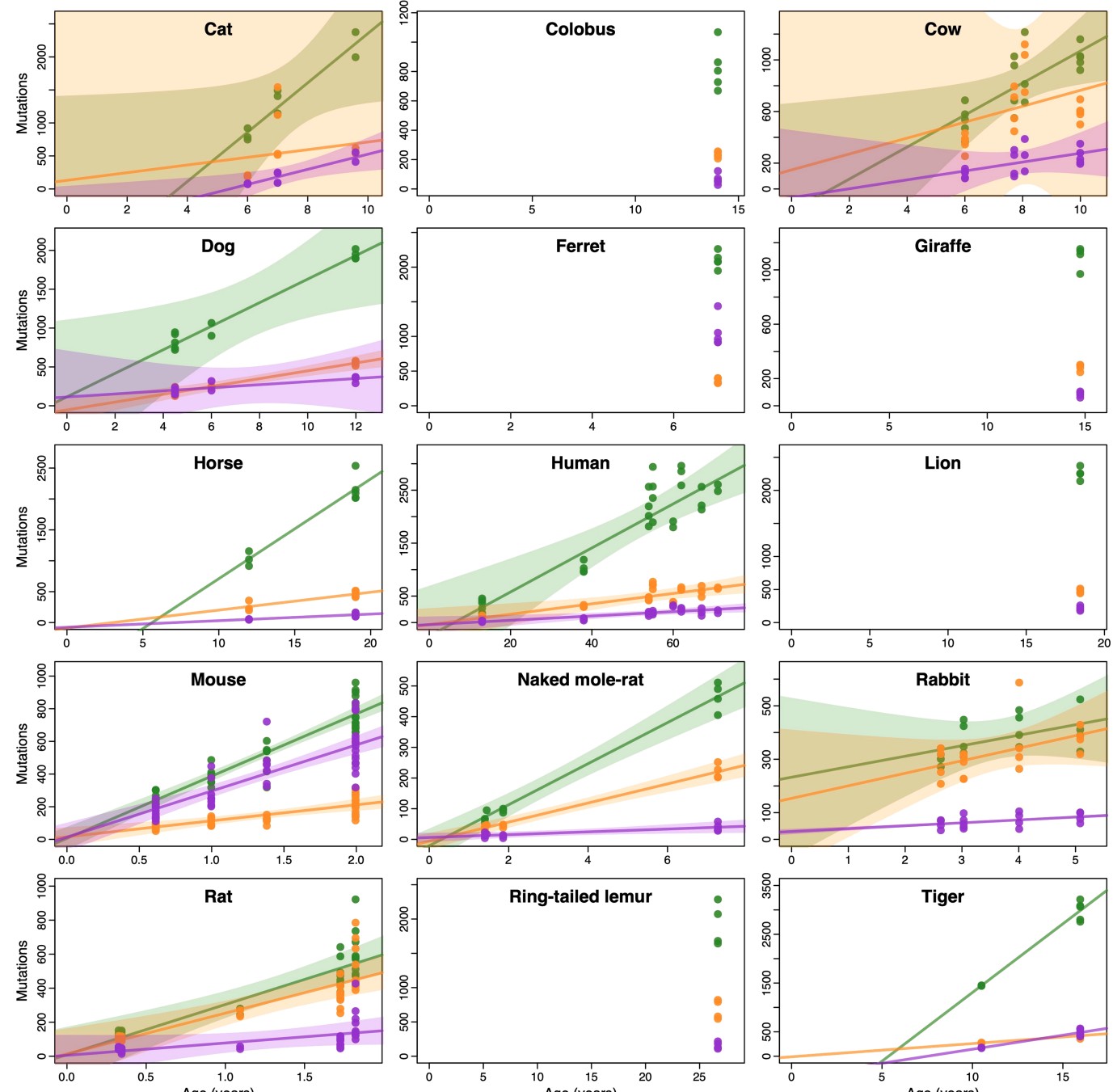

**Extended Data Fig. 5 | Signature-specific mutation accumulation across species.** Each panel presents somatic substitution burdens per genome for mutational signatures SBS1 (green), SBSB (yellow) and SBSC (purple) in a given species. For species with two or more individuals, the estimated regression lines from simple linear regression models on individual mean burdens per signature are shown. For species with three or more individuals, shaded regions indicate 95% confidence intervals of the regression lines. Harbour porpoise samples were excluded owing to unknown age of the sampled individual.

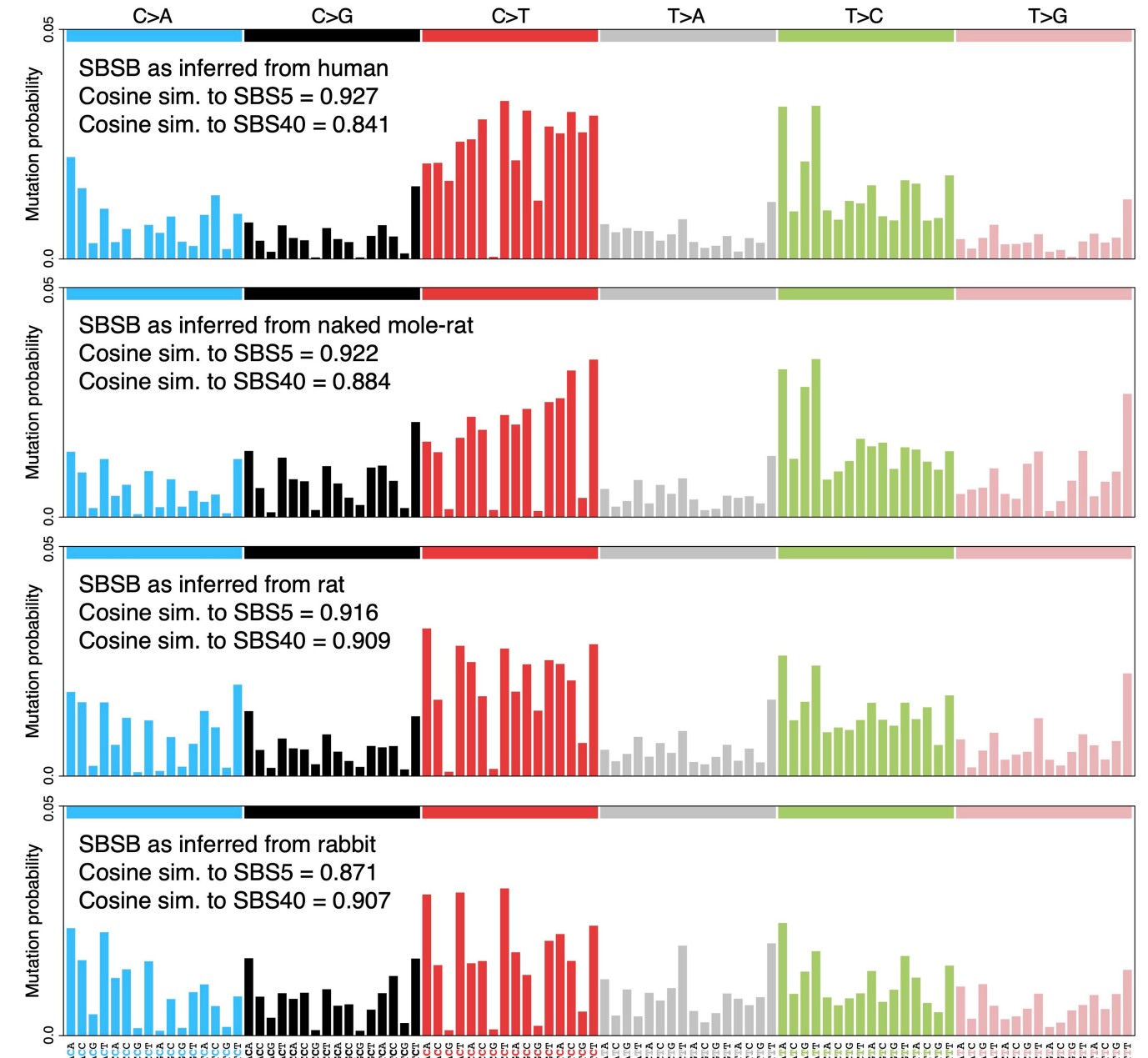

**Extended Data Fig. 6 | Profiles of signature SBSB as inferred from different species.** Trinucleotide-context mutational spectra of signature SBSB, as inferred independently from somatic mutations in crypts from four representative species (top to bottom): human, naked mole-rat, rat and rabbit (Methods). Signatures are shown in a human-genome-relative representation. Cosine similarities between each signature and the COSMIC human signatures SBS5 and SBS40 are provided.

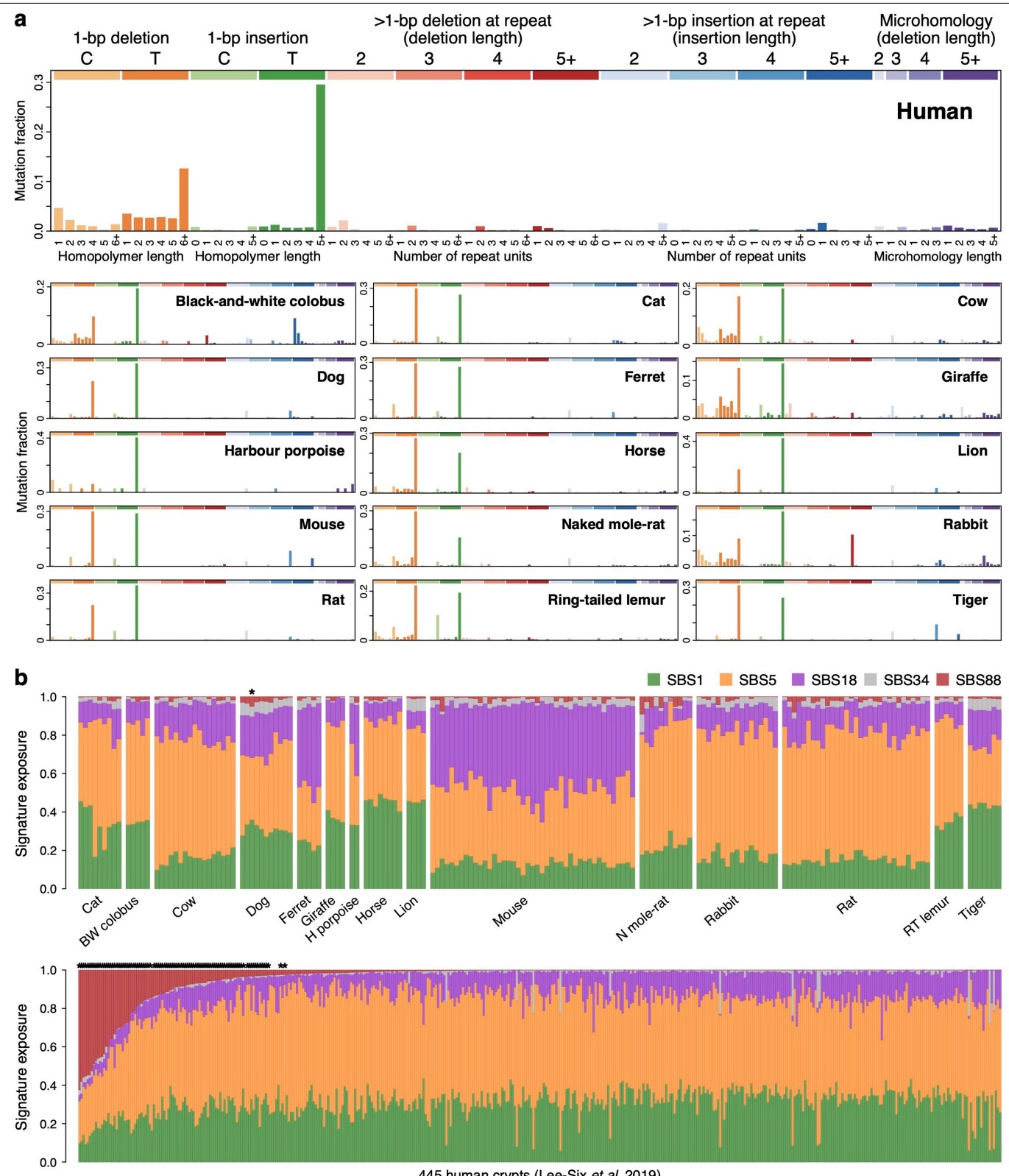

**Extended Data Fig. 7 | Somatic indels and colibactin exposure. a**, Mutational spectra of somatic indels in each species. The *x* axis shows 83 types of insertion or deletion, coloured by type and length[30]. **b**, Colibactin exposure in non-human and human colorectal crypts. Exposures to mutational signatures SBS1, SBS5, SBS18, SBS34 and SBS88, as inferred by expectation–maximization, for 180 non-human crypts in this study (top) and 445 human crypts sequenced in a previous study[8]. Asterisks indicate samples with statistically significant colibactin (SBS88) exposure, based on a LRT (Methods). BW, black-and-white; H, harbour; N, naked; RT, ring-tailed.

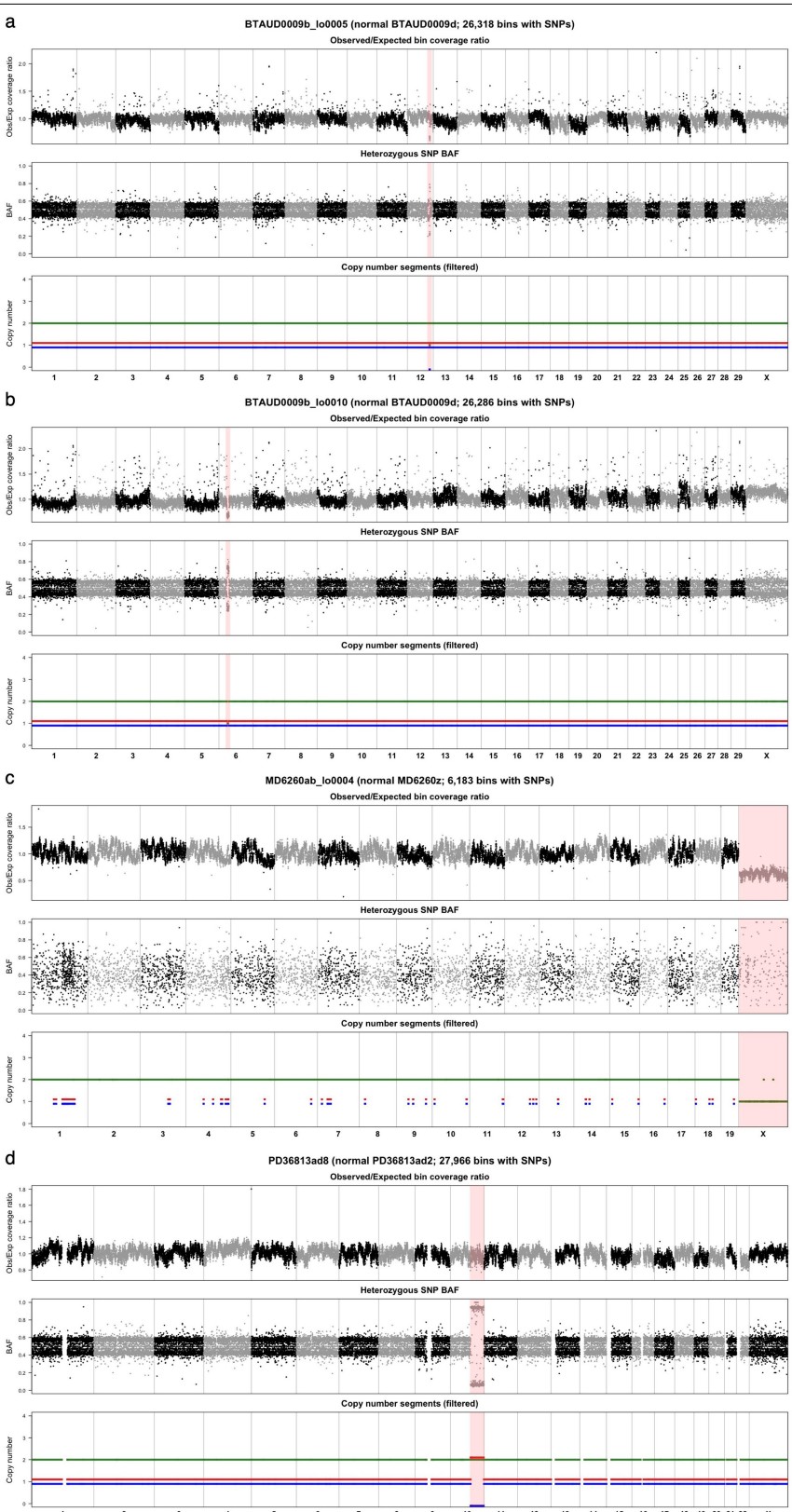

**Extended Data Fig. 8 | Identified copy number changes. a–d,** Somatic copy number changes in cow (**a**, **b**), mouse (**c**) and human (**d**) colorectal crypts. In each case, chromosomes are presented along the *x* axis, with each point representing a 100-kb genomic bin. The top panel presents the ratio between observed and expected sequencing coverage per bin; the middle panel shows the median BAF of heterozygous germline SNPs per bin; and the bottom panel presents the inferred segments of total copy number (green) and allele-specific copy number (red/blue). Regions of copy number change are highlighted in pink. The sparsity of BAF and allele-specific copy number values in the mouse crypt (**c**) are related to the fact that mouse samples generally had very low numbers of germline SNPs.

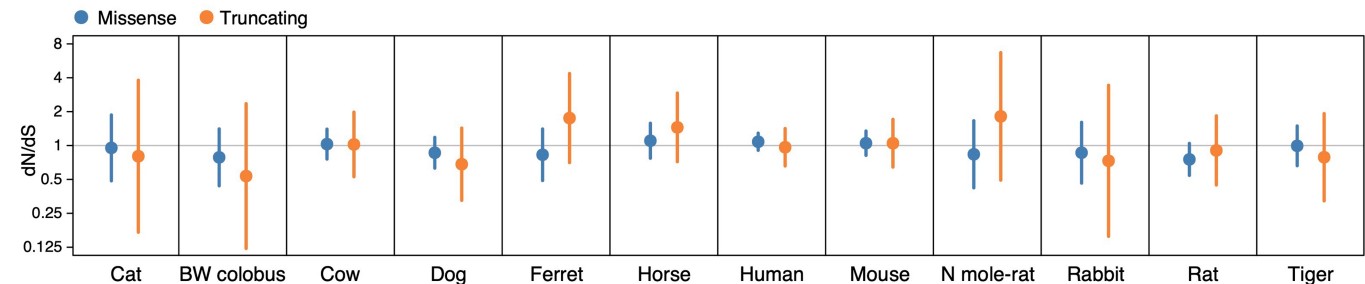

**Extended Data Fig. 9 | Somatic dN/dS.** Estimates of dN/dS for missense and truncating somatic mutations in each of the species with available genome annotation. Dots and error bars represent maximum likelihood estimates and 95% confidence intervals, respectively ($n$ = 27, 2, 32, 2, 136, 12, 118, 9, 39, 7, 102, 10, 440, 34, 231, 22, 25, 3, 30, 2, 110, 10, 75 and 6 mutations, left to right). Note the logarithmic scale of the $y$ axis.

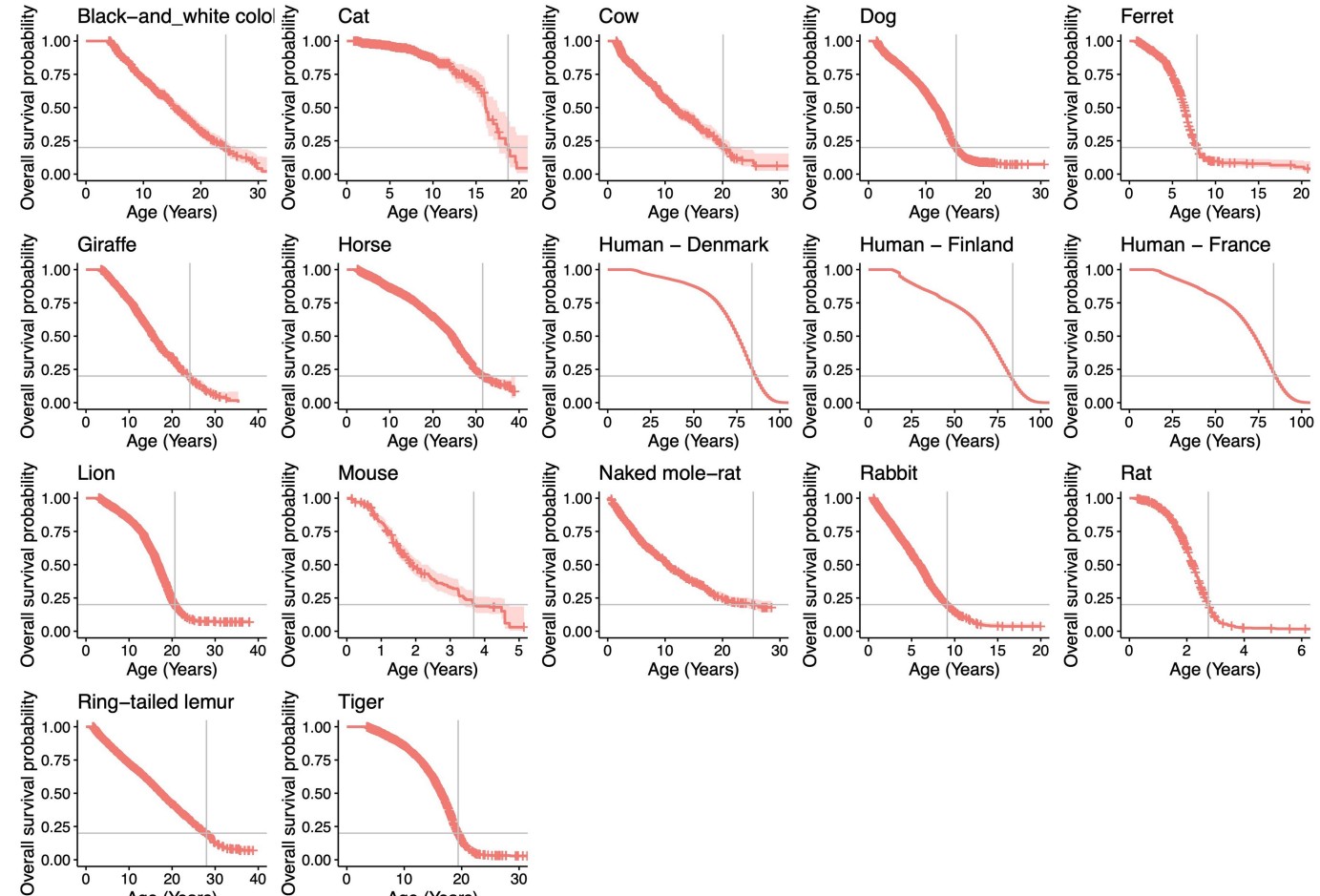

**Extended Data Fig. 10 | Kaplan–Meier curves of longevity in captivity.** Kaplan–Meier survival curves for each species, calculated using captive lifespan data from Species360 for non-human species and census record data for humans (Methods). The shaded areas represent 95% confidence intervals of the survival curves. A horizontal grey bar indicates the age at which 80% of individuals had already died (80th percentile), which was adopted as a robust estimate of species lifespan.

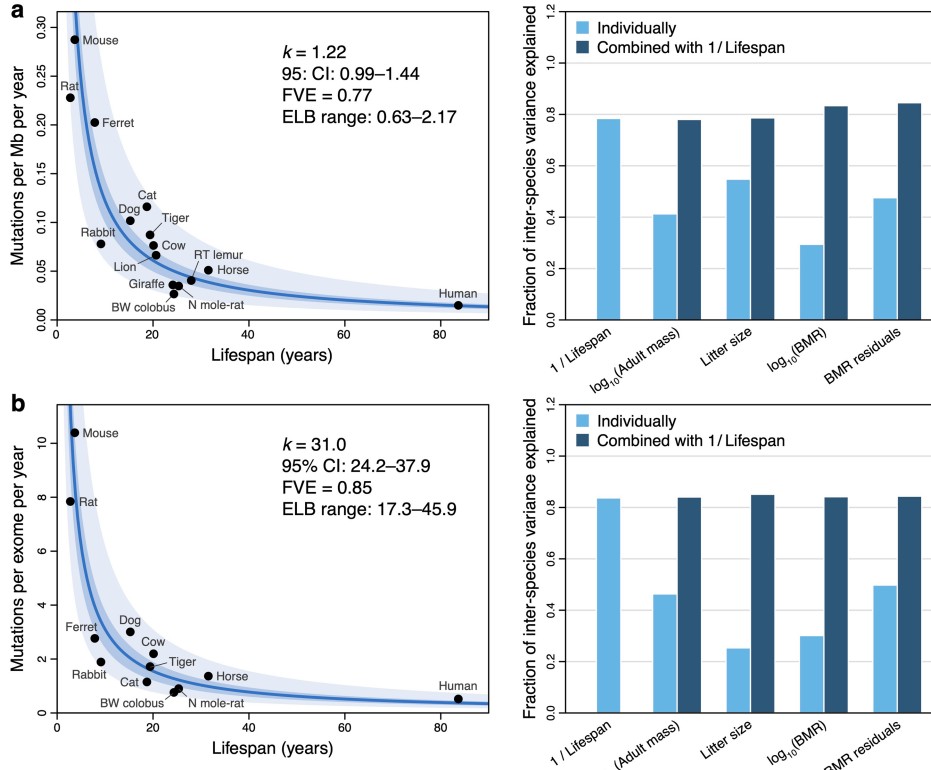

**Extended Data Fig. 11 | Associations between life-history variables and alternative measures of somatic mutation rate. a, b,** Same analyses as Fig. 3c, f, but using somatic mutation rates per megabase (**a**), or per protein-coding exome (**b**), rather than per genome (Methods). Leftmost panels show zero-intercept LME regressions of somatic mutation rates on inverse lifespan (1/lifespan), presented on the scale of untransformed lifespan ($x$ axis). The $y$ axes present mean mutation rates per species, although mutation rates per crypt were used in the regressions. Darker shaded areas indicate 95% confidence intervals (CI) of the regression lines; lighter shaded areas mark a two-fold deviation from the regression lines. Point estimate and 95% CI of the regression slope ($k$), fraction of inter-species variance explained (FVE), and range of ELB are provided. Rightmost panels show comparisons of FVE values achieved by free-intercept LME models using inverse lifespan and other life-history variables (alone or in combination with inverse lifespan) as explanatory variables. BW, black-and-white; N, naked; RT, ring-tailed.

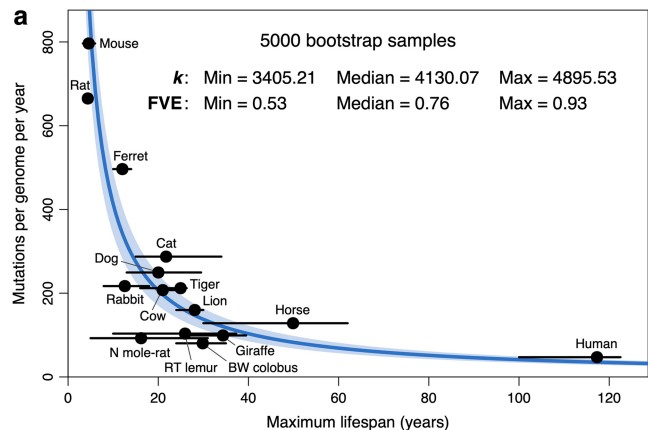

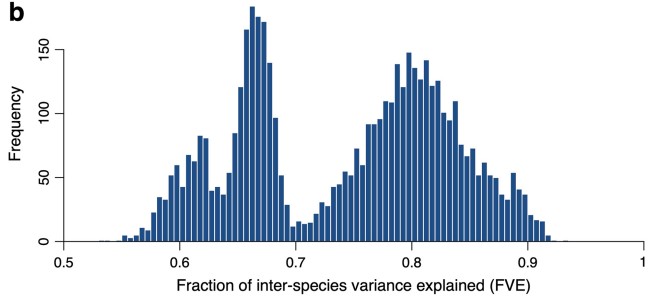

**Extended Data Fig. 12 | Bootstrapped regression of somatic mutation rates on published lifespan estimates. a**, Bootstrapped regression of somatic substitution rates on the inverse of lifespan (1/lifespan), using a zero-intercept LME model (Methods). For each of 5,000 bootstrap samples (replicates), lifespan values per species were randomly chosen from a set of published maximum longevity estimates (Supplementary Table 6). The blue line indicates the median regression slope (*k*) across bootstrap samples, and the shaded area depicts the range of estimates of *k* across bootstrap samples. Black dots and error bars indicate the mean and range, respectively, of published longevity estimates for each species. The median and range of both *k* and the fraction of inter-species variance explained (FVE) are provided. **b**, Histogram of FVE values across the 5,000 bootstrap samples.

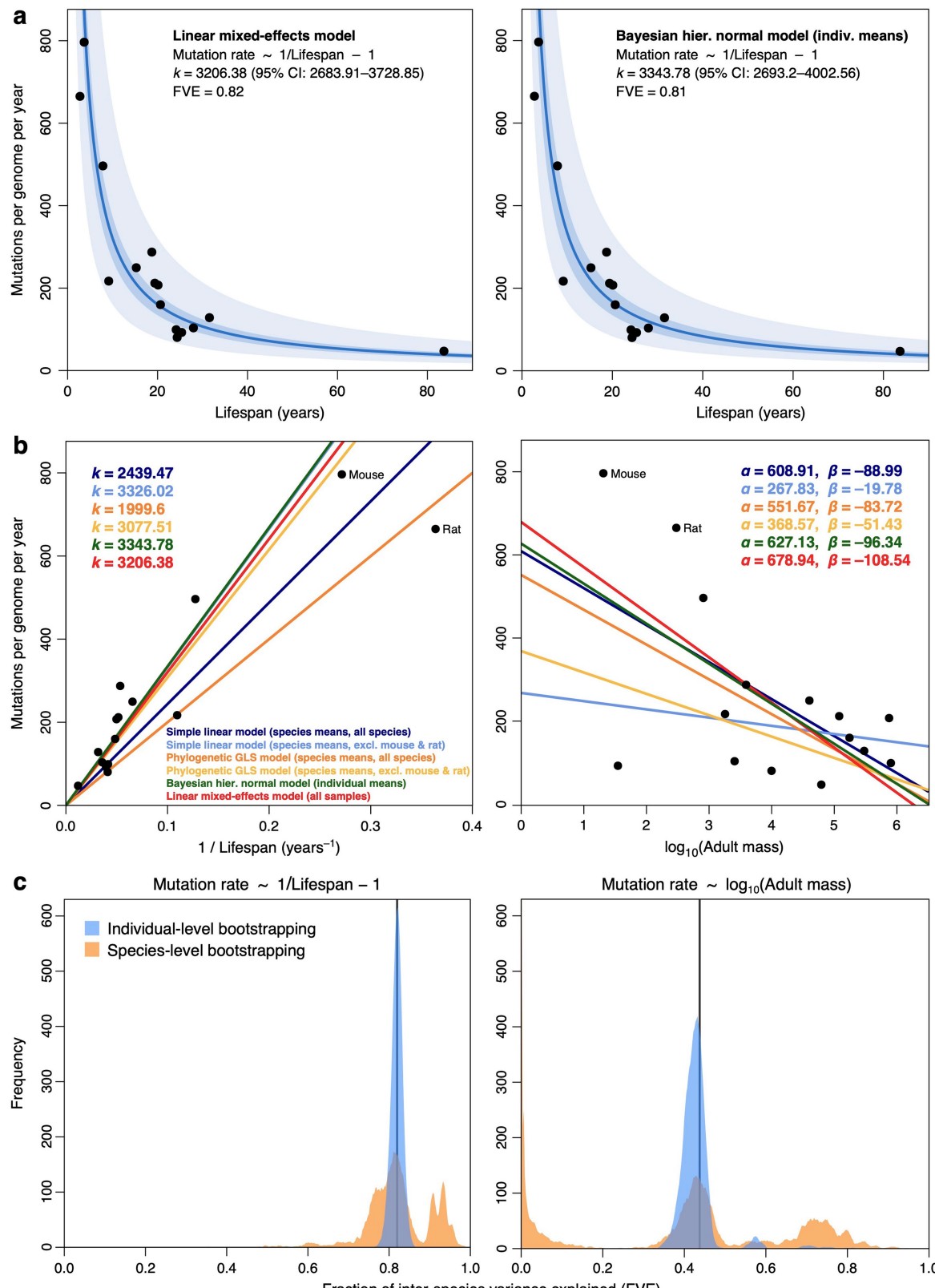

**Extended Data Fig. 13** | See next page for caption.

**Extended Data Fig. 13 | Comparison of regression models for somatic mutation rates. a**, Zero-intercept regression of somatic substitution rates on inverse lifespan (1/lifespan), using a LME model applied to mutation rates per crypt (left) and a Bayesian hierarchical normal regression model applied to mean mutation rates per individual (Methods). For simplicity, black dots present mean mutation rates per species. Darker shaded areas indicate 95% confidence/credible intervals (CI) of the regression lines; lighter shaded areas mark a two-fold deviation from the regression lines. Point estimates and 95% CI of the regression slopes ($k$) and fraction of inter-species variance explained (FVE) are provided. **b**, Comparison of regression lines for the regression of somatic substitution rates on 1/lifespan (left; zero intercept) and log-transformed adult body mass (right; free intercept), using simple linear models (dark and light blue), phylogenetic generalized least-squares models (orange and yellow), Bayesian hierarchical normal models (green) and LME models (red) (Methods). Point estimates of the regression coefficients for each model are provided. **c**, Distributions of regression FVE under individual- and species-level bootstrapping. For the LME models regressing somatic mutation rates on inverse lifespan (zero intercept; left) and log-transformed mass (free intercept), the curves present distributions of FVE from 10,000 bootstrap replicates, obtained through random resampling of either individuals (blue) or species (orange) (Methods). Vertical lines indicate the FVE values obtained using the entire dataset.

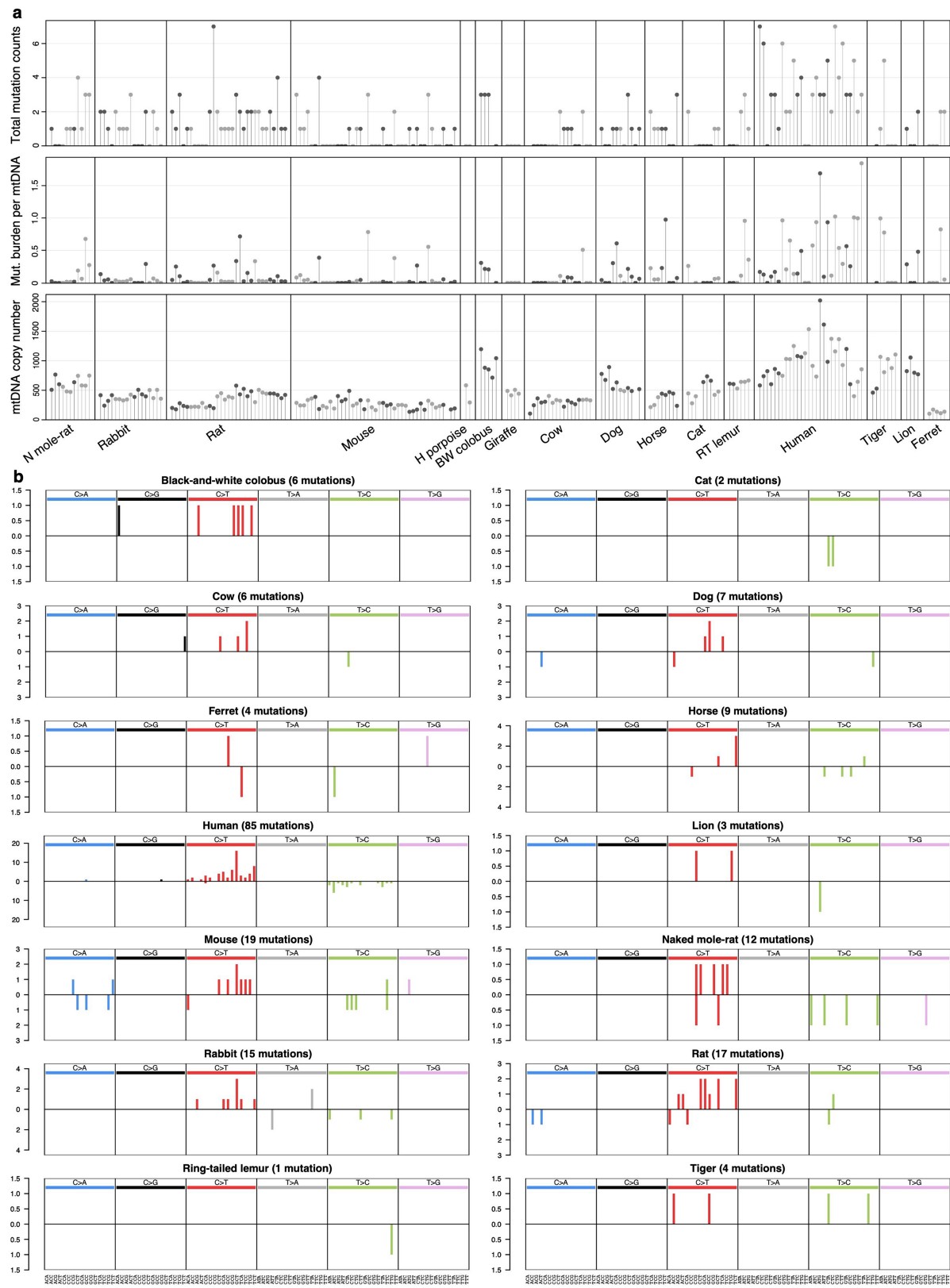

**Extended Data Fig. 14 | mtDNA mutation burden and spectrum. a**, Total somatic mtDNA mutations called (substitutions and indels; top), somatic mutation burden per mtDNA copy (middle), and estimated mtDNA copy number in each crypt sample. Samples are arranged horizontally as in Fig. 1b, with samples from the same individual coloured in the same shade of grey.

**b**, Mutational spectra of mtDNA substitutions in each species. The *x* axis shows 96 mutation types on a trinucleotide context, coloured by base substitution type; the *y* axis shows mutation counts. Mutations on the upper and lower halves of the spectrum represent substitutions with the pyrimidine base located on the heavy and light strands of mtDNA, respectively.

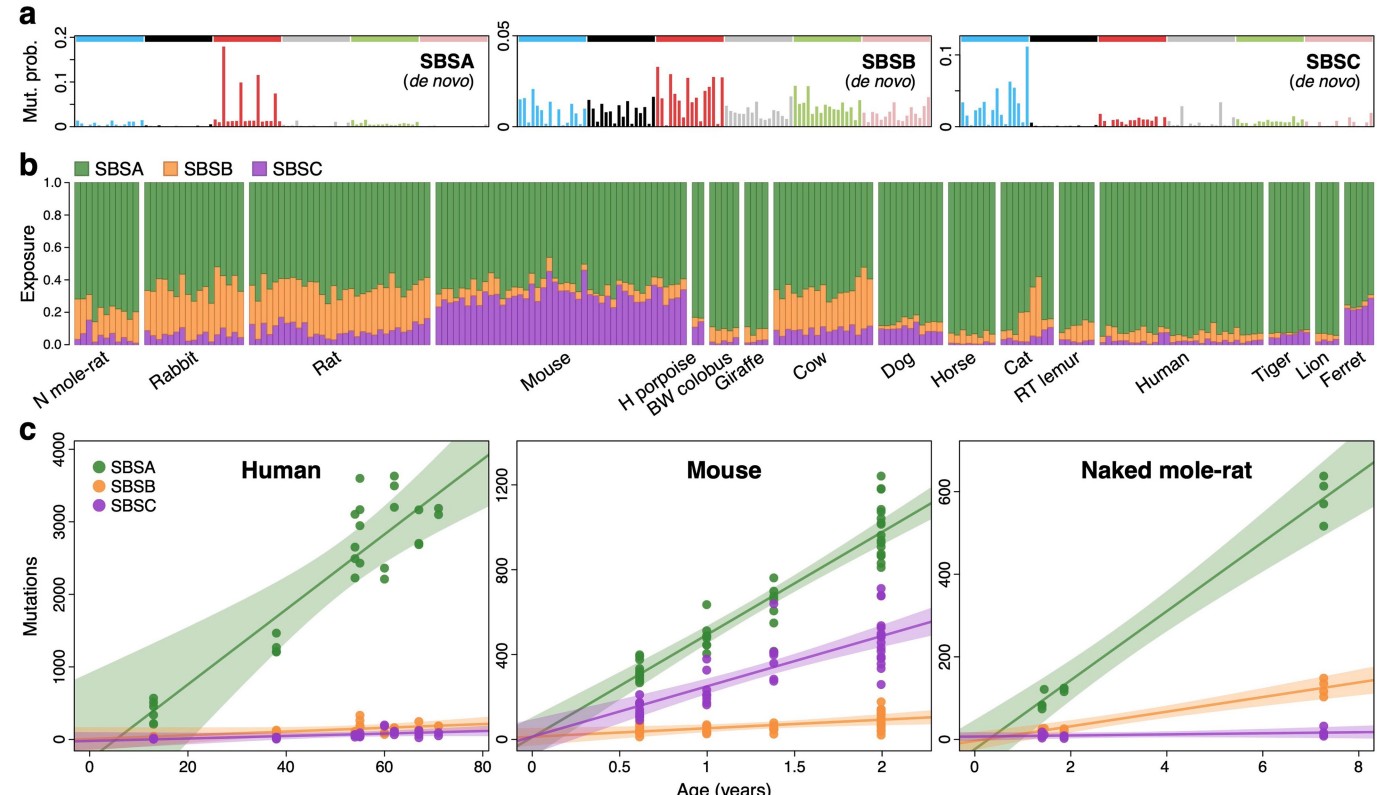

**Extended Data Fig. 15 | Mutational signatures and exposures as inferred de novo. a**, Mutational signatures inferred de novo from the species mutational spectra shown in Fig. 2a. Signatures are shown in a human-genome-relative representation. SBSA is the de novo equivalent of COSMIC signature SBS1 (Fig. 2b). **b**, Exposure of each sample to each of the mutational signatures shown in **a**. Samples are arranged horizontally as in

Fig. 1b. **c**, Regression of signature-specific mutation burdens on individual age for human, mouse and naked mole-rat samples. Regression was performed using mean mutation burden per individual. Shaded areas indicate 95% confidence intervals of the regression lines. BW, black-and-white; H, harbour; N, naked; RT, ring-tailed.

# Reporting Summary

Nature Research wishes to improve the reproducibility of the work that we publish. This form provides structure for consistency and transparency in reporting. For further information on Nature Research policies, see our Editorial Policies and the Editorial Policy Checklist.

## Statistics

For all statistical analyses, confirm that the following items are present in the figure legend, table legend, main text, or Methods section.

| n/a | Confirmed | |
|---|---|---|
| ☐ | ☒ | The exact sample size (*n*) for each experimental group/condition, given as a discrete number and unit of measurement |
| ☐ | ☒ | A statement on whether measurements were taken from distinct samples or whether the same sample was measured repeatedly |
| ☐ | ☒ | The statistical test(s) used AND whether they are one- or two-sided<br>*Only common tests should be described solely by name; describe more complex techniques in the Methods section.* |
| ☐ | ☒ | A description of all covariates tested |
| ☐ | ☒ | A description of any assumptions or corrections, such as tests of normality and adjustment for multiple comparisons |
| ☐ | ☒ | A full description of the statistical parameters including central tendency (e.g. means) or other basic estimates (e.g. regression coefficient) AND variation (e.g. standard deviation) or associated estimates of uncertainty (e.g. confidence intervals) |
| ☐ | ☒ | For null hypothesis testing, the test statistic (e.g. *F*, *t*, *r*) with confidence intervals, effect sizes, degrees of freedom and *P* value noted<br>*Give P values as exact values whenever suitable.* |
| ☐ | ☒ | For Bayesian analysis, information on the choice of priors and Markov chain Monte Carlo settings |
| ☐ | ☒ | For hierarchical and complex designs, identification of the appropriate level for tests and full reporting of outcomes |
| ☒ | ☐ | Estimates of effect sizes (e.g. Cohen's *d*, Pearson's *r*), indicating how they were calculated |

*Our web collection on statistics for biologists contains articles on many of the points above.*

## Software and code

Policy information about availability of computer code

| Data collection | No software was used |
|---|---|
| Data analysis | Most analyses have been done with bespoke pipelines, deposited in Zenodo (https://doi.org/10.5281/zenodo.5554801) and GitHub (https://github.com/baezortega/CrossSpecies2021). Analyses in R were done with R v3.6.2. R packages used include: caper (v1.0.1), deepSNV (v1.32.0), dNdScv (v0.0.1.0), MutationalPatterns (v1.12.0), nlme (v3.1-137), sigfit (v2.1.0). Our pipeline makes use of the software BWA (v0.7.17-r1188), CaVEMan (v1.13.15), Pindel (v3.3.0), bedtools (v2.17.0), biobambam2 (v2.0.86), Indelwald (v24/09/2021). |

For manuscripts utilizing custom algorithms or software that are central to the research but not yet described in published literature, software must be made available to editors and reviewers. We strongly encourage code deposition in a community repository (e.g. GitHub). See the Nature Research guidelines for submitting code & software for further information.

## Data

Policy information about availability of data

All manuscripts must include a data availability statement. This statement should provide the following information, where applicable:
- Accession codes, unique identifiers, or web links for publicly available datasets
- A list of figures that have associated raw data
- A description of any restrictions on data availability

DNA sequence data have been deposited in the European Genome-Phenome Archive (EGA; https://ega-archive.org) under overarching accession EGAD00001008032. Preprocessed data files used in the analyses have been deposited in Zenodo (https://doi.org/10.5281/zenodo.5554777). Human DNA sequence data from a previous study (Lee-Six et al., 2019) are deposited in EGA (accession EGAD00001004192).

# Field-specific reporting

Please select the one below that is the best fit for your research. If you are not sure, read the appropriate sections before making your selection.

☒ Life sciences  ☐ Behavioural & social sciences  ☐ Ecological, evolutionary & environmental sciences

For a reference copy of the document with all sections, see nature.com/documents/nr-reporting-summary-flat.pdf

# Life sciences study design

All studies must disclose on these points even when the disclosure is negative.

| | |
|---|---|
| Sample size | Information on sample sizes is provided for all analyses. We selected samples from available individuals attempting to span a wide range of ages. The requested sequencing coverage (40x) was chosen to achieve high sensitivity and specificity for clonal somatic variants. |
| Data exclusions | We excluded 41 samples due to evidence of polyclonality or poor sequencing quality. The criteria used to assess sample clonality and quality are explained in the Methods, 'Sample filtering' section. |
| Replication | To confirm the reproducibility of somatic variant calls we used laser capture microdissection to microdissect and sequence two sections from the same mouse colorectal crypt. Both sections were submitted for independent library preparation, genome sequencing, variant calling and filtering using our pipeline. The vast majority of somatic substitution calls were shared between both sections (see Methods & Supplementary Figure 1c), confirming the replicability of our somatic variant calls.<br>Mutation signature extraction was performed with two different methods that gave broadly consistent results (Methods).<br>The main regression results were replicated using a number of different regression models (Methods). |
| Randomization | Our study design did not involve experimental groups. Covariates were controlled for by including them in our regression models (Methods). |
| Blinding | We did not apply randomization because we did not have a case/control study design or treatment groups. While sample metadata (such as animal age) did not inform the variant calling process, which was applied in an identical manner for all samples, there was no enforced blinding procedure. |

# Reporting for specific materials, systems and methods

We require information from authors about some types of materials, experimental systems and methods used in many studies. Here, indicate whether each material, system or method listed is relevant to your study. If you are not sure if a list item applies to your research, read the appropriate section before selecting a response.

### Materials & experimental systems

| n/a | Involved in the study |
|---|---|
| ☒ | ☐ Antibodies |
| ☒ | ☐ Eukaryotic cell lines |
| ☒ | ☐ Palaeontology and archaeology |
| ☐ | ☒ Animals and other organisms |
| ☒ | ☐ Human research participants |
| ☒ | ☐ Clinical data |
| ☒ | ☐ Dual use research of concern |

### Methods

| n/a | Involved in the study |
|---|---|
| ☒ | ☐ ChIP-seq |
| ☒ | ☐ Flow cytometry |
| ☒ | ☐ MRI-based neuroimaging |

## Animals and other organisms

Policy information about studies involving animals; ARRIVE guidelines recommended for reporting animal research

| | |
|---|---|
| Laboratory animals | We did not maintain laboratory animals specifically for this study. Tissue samples of rabbit were purchased from a commercial provider. Samples from mouse, rat and naked mole rat were obtained from collaborators maintaining these lines for other research projects. Samples from other species were collected opportunistically at necropsy. The species, strains, individuals, age and source are reported in extended data tables 1 and 4. |
| Wild animals | Sample materials were collected from a stranded wild harbour porpoise by the UK Cetacean Strandings Investigation Program (CSIP). The individual was deceased at the time of sample collection. CSIP is funded by Defra and the devolved administration to investigate and document strandings of cetaceans and other marine life around the UK coastline. |
| Field-collected samples | The study did not involve animals collected from the field. |
| Ethics oversight | All animal samples were obtained with the approval of the local ethical review committee (AWERB) at the Wellcome Sanger Institute and those at the holding institutions. |

Note that full information on the approval of the study protocol must also be provided in the manuscript.

