## [Peer Review File · Nature]

Manuscript Title: Somatic mutation rates scale with lifespan across mammals

Reviewer Comments & Author Rebuttals

Reviewer Reports on the Initial Version:

Referees' comments:

Referee #1 (Remarks to the Author):

In their manuscript entitled “Somatic mutation rates scale with lifespan across mammals”, Cagan et al. use whole genome sequencing of individual colonic crypts to assess somatic mutation patterns across 16 mammalian species. Central questions motivating this study are “Peto’s paradox”, the lack of association between body mass/lifespan and cancer risk across species, and the role of somatic mutagenesis in aging. By extracting mutational signatures from the mutation data in each species, the authors detect only 3 signatures, which closely resemble human SBS1, SBS5, and SBS18. All 3 signatures linearly correlate with age in each species, although the relative contributions and associations of the signatures with age vary across species. The authors’ most striking finding is a strong inverse correlation between mutation rate per year and lifespan across the 16 species. The correlation between mutation rate and lifespan appears to be stronger than the correlation with other life-history variables such as body mass or basal metabolic rate.

General comments:

This paper is straight-forward and extremely carefully conceived. Its thoughtful experimental design and novel central conclusion leave little room for substantial improvements. However, in my opinion, the manuscript would benefit from deeper statistical analyses of the robustness of the 1/lifespan relationship. Acknowledging the herculean effort of assembling this dataset, given the small fraction of mammalian species represented and small number of individuals profiled, I think it is important to not overstate the certainty with which we can say that 1/lifespan is the “true” relationship underlying the data. The authors make this point fairly strongly on the basis of relatively limited data. I am not sure I quite understand why – in my opinion, the value or novelty of this work does not hinge on this point and a slightly more careful treatment of the matter will not take away from the story.

Specific comments:

1. To what degree is the superior correlation between mutation rate and 1/lifespan (compared to the other life-history relationships) a property of this specific dataset? We only have 16 data points, each measured with quite a bit of uncertainty, and very few species at the ends of the age distribution spectrum (mouse, rat, ferret, human). Can we somehow convince ourselves that lifespan did not win the “fit contest” by chance? For example by re-sampling the available data in creative ways, can the authors show the stability of this claim?
2. More philosophically speaking, is there a strong reason to believe that only one of the observed covariates (lifespan, body mass, BMR etc.) constitutes the “true” explanation for the mutation rate

variation? Could they not all contribute, perhaps even to different degrees in the evolution of different species? Perhaps the authors can articulate their underlying thinking a bit more clearly on this point.

3. Can a similar analysis as described in point 1 be done for the zero intercept claim for those species where multiple individuals were sequenced (Figure 1c)?

4. Can confidence intervals be added to the bars in Fig 3d, e.g. by bootstrapping?

5. It is interesting that rat and mouse have similar SBS1 rates, similar sizes, and similar lifespans, and yet rat has the most extreme SBSB rate, and mouse the most extreme SBSC rate. Can the authors hypothesize what might drive this difference? What mechanistic hypotheses could explain the different contributions of SBS1, SBSB and SBSC across the different species?

6. There are two entries for Human in ED Table 7.

7. I apologize if I missed this in the methods, but are the authors worried that mutation calling in samples that are not entirely monoclonal (e.g. harbor porpoise, some of the horse and tiger samples) might be problematic? Might we be overestimating mutation burden by counting mutations from multiple crypts or missing some due to reduced sensitivity?

8. The discussion might benefit from including some more controversial points. Currently the narrative is almost a little bit too “optimized” and smooth (at least for my taste). In their discussion, the authors write “The most striking finding of this study is the inverse scaling of somatic mutation rates with lifespan. This has been a long-standing prediction of the somatic mutation theory of ageing. The observation is consistent with a causative role of somatic mutations in mammalian ageing.” But some of the authors of this paper have recently published two Biorxiv preprints showing that individuals with increased mutation rates due to deficient polymerases or DNA mismatch repair do not show accelerated aging (only an increased risk of cancer). From the abstract of one of those Biorxiv papers: “The results, therefore, do not support a simple model in which all features of ageing are attributable to widespread cell malfunction directly resulting from somatic mutation burdens accrued during life.” How do the authors reconcile these findings with their speculation that mutation rates are a primary cause of aging?

9. Also in the discussion, can the authors speculate on how their findings are expected to relate to cancer risk? Can they include some thoughts about the number of cells that undergo the described mutagenesis in each species and how this number may relate to organism-level cancer risk? Peto’s paradox refers partially to lack of association between the number of cells in an organism and cancer risk. Even if individual end-of-life cells across species have a fairly similar mutation burden, overall cancer risk might still be expected to scale with the number of cells in an organism. Of course, we know that it does not, but I think it would be important to clarify that this work does not “solve” Peto’s paradox (nor does it need to – that would be a very high bar).

Referee #2 (Remarks to the Author):

Cagan et al. asked whether somatic mutation rates scale across a selection of mammals in relation to the measured species-specific lifespan (80% max scored survivorship). Remarkably, they find that species-specific mutation rate inversely correlates with species-specific lifespan (short-lived species have higher mutation rates) and that the life-long cumulative mutation burden is comparable across species. Their analysis is rigorous, convincing and their conclusions are fully supported by their results.

The authors measured somatic mutation rates in intestinal crypts across a spectrum of mammals, which differ in body size, lifespan, metabolic rates and phylogenetic relatedness. They call somatic variants in intestinal crypts against skin, using a sequencing depth of 30x. They compute single-base-substitutions and indels in nuclear and mitochondrial genomes across all species.

I congratulate the authors for a great piece of work. I greatly enjoyed reading this manuscript, which I consider a milestone and a breath of pure oxygen in the field of comparative genomics of aging.

What this work shows is a scale-free relationship between the rate of somatic mutations and species-specific lifespan. While the mechanistic part related with the mutational spectra across species alludes to specific shared mechanisms responsible for mutation rates to scale across species with lifespan - and this could have deserved further investigation and mechanistic analysis - the novelty and solidity of the main finding in my opinion is worthy of publication on nature.

Questions and comments:

1. Overall, the quality of the figures is not "impressive" and rather cheap. I find that more effort could be placed in making figures 1b and 3a more intuitive to the readers.
2. Have the authors "controlled" somatic mutation rate by estimated genome size (e.g. considered as an explanatory variable)? In several species, (germline, not somatic) mutation rate significantly scale with genome size (e.g., check Cui et al. <https://doi.org/10.1016/j.cell.2019.06.004>).
3. How far in phylogeny does the relationship between mutation rate and lifespan scale? What about century-long lived whales? What about super long-lived bats? What about beyond mammals? It would be nice if the authors could at least comment on this.
4. I understand that the intercept for the mutation rate model is negligible, as represented in figure 1c. This finding justifies the use of a zero-intercept mixed effect model. However, it would be important to know what happens with a multilevel mixed effect model, where intercept and slope are "learned" from the data. This way, the authors can report an actual mean value for the intercepts in their model.

Referee #3 (Remarks to the Author):

In their study, the authors report a detailed survey of somatic mutation rates in 16 species. I am not aware of a previous attempt to measure somatic mutation rates in so many species and with these methods or others that would provide such a detailed comparative portrait of somatic mutation. Their findings were that somatic mutation rate scales best with lifespan rather than mass or other life history traits and that mutational patterns are largely the same across these mammals. They also report that despite 30-fold range in lifespan, the end of life mutational burden only ranges about 3 fold across these species. I found the approach to be powerful and the conclusions to be impactful.

While these specific methods are not my specialty, the authors convincingly argue that their model (the intestinal crypt) and methods (resequencing of multiple samples and individuals with detailed filtering) are sound and ideal for these questions. key diagnostic results are also in-line with previously reported conclusions from human studies.

The analyses were also sound in my view and yet simple enough to be general and robust and understood by most readers. For example, the amount of variance in mutation rate explained by

inverse age was not subtle (82%) and this speaks to the strength of that conclusion.

I find the study to be exciting, clear, well written and well executed. Proper controls were performed and alternative explanations were explored. From my perspective, I did not find many concerns. I would like to see this manuscript published at Nature.

Minor Concern:

1. It would help the reader to have more explanation of how mutational signatures are derived and what they represent, when they are first presented.

Author Rebuttals to Initial Comments:

Referee expertise:

Referee #1: cancer genomics, somatic evolution

Referee #2: evolutionary genetics of ageing

Referee #3: comparative genomics

Referees' comments:

Referee #1 (Remarks to the Author):

Reviewer: In their manuscript entitled “Somatic mutation rates scale with lifespan across mammals”, Cagan et al. use whole genome sequencing of individual colonic crypts to assess somatic mutation patterns across 16 mammalian species. Central questions motivating this study are “Peto’s paradox”, the lack of association between body mass/lifespan and cancer risk across species, and the role of somatic mutagenesis in aging. By extracting mutational signatures from the mutation data in each species, the authors detect only 3 signatures, which closely resemble human SBS1, SBS5, and SBS18. All 3 signatures linearly correlate with age in each species, although the relative contributions and associations of the signatures with age vary across species. The authors’ most striking finding is a strong inverse correlation between mutation rate per year and lifespan across the 16 species. The correlation between mutation rate and lifespan appears to be stronger than the correlation with other life-history variables such as body mass or basal metabolic rate.

General comments:

Reviewer: This paper is straight-forward and extremely carefully conceived. Its thoughtful experimental design and novel central conclusion leave little room for substantial improvements. However, in my opinion, the manuscript would benefit from deeper statistical analyses of the robustness of the 1/lifespan relationship. Acknowledging the herculean effort of assembling this dataset, given the small fraction of mammalian species represented and small number of individuals profiled, I think it is important to not overstate the certainty with which we can say that 1/lifespan is the “true” relationship underlying the data. The authors make this point fairly strongly on the basis of relatively limited data. I am not sure I quite understand why – in my opinion, the value

or novelty of this work does not hinge on this point and a slightly more careful treatment of the matter will not take away from the story.

Specific comments:

Reviewer: 1. To what degree is the superior correlation between mutation rate and 1/lifespan (compared to the other life-history relationships) a property of this specific dataset? We only have 16 data points, each measured with quite a bit of uncertainty, and very few species at the ends of the age distribution spectrum (mouse, rat, ferret, human). Can we somehow convince ourselves that lifespan did not win the “fit contest” by chance? For example by re-sampling the available data in creative ways, can the authors show the stability of this claim?

Authors' response: We thank the reviewer for this suggestion.

To shorten the main text, we had originally described several supportive statistical analyses only in the Methods. Following the reviewer's question, we have strengthened this part of the main text with additional statistical analyses and 4 additional panels in Fig 3. All of the analyses support the original conclusions.

The analyses are described in more detail below and include:

1. An allometric regression of lifespan before introducing the 1/lifespan model. The allometric slope is not significantly different from -1, supporting the simpler and theoretically more meaningful 1/lifespan model.
2. Throughout the paper, we used linear mixed effect (LME) regression models to perform robust inferences on these data. Whereas it is true that we only have data from 16 species, LME regression models enable us to utilise the information from 208 genomes and 56 individuals. This makes better use of the full dataset than simple re-sampling. However, we have also added a bootstrap analysis as suggested by the reviewer.
3. To assess whether body size or other life history variables explain some of the variation in somatic mutation rates across species independently of lifespan, we perform two separate analyses: LME nested models and partial correlation analysis.

1. Allometric regression on lifespan

The 1/lifespan model is one of the simplest possible relationships between somatic mutation rates and lifespan and a long-standing theoretical prediction. If all species end

their lifespans with the same mutation burden per cell, then somatic mutation rates will vary according to $1/\text{lifespan}$ ($\text{rate} \times \text{lifespan} = k$, which can be rewritten as $\text{rate} = k \times 1/\text{lifespan}$, where k is the end-of-lifespan mutation burden). k can then be inferred from a zero-intercept linear regression using a $1/\text{lifespan}$ transformation (a simple 1-parameter regression).

This 1-parameter regression was the original analysis presented in the main text. However, this was also supported by an unconstrained allometric regression and by alternative analyses of the data, which were only shown in the Methods. With r being the somatic mutation rate per cell per year and L being the lifespan of a species, an allometric relationship between them can be expressed as: $r = k L^a$. Using the usual log conversion, this yields $\log(r) = \log(k) + a \log(L)$. This model can then be fitted using a standard linear regression with 2 parameters (intercept and slope). Applying this to lifespan yielded a linear relationship with a slope ($=a$) close to and not significantly different from -1, further supporting the use of the simpler $1/\text{lifespan}$ model: $r = k L^{-1} = k (1/L)$. This analysis clarifies that the choice of $1/\text{lifespan}$ was data-driven and not merely theory-driven, but we did not include it in the original main text. This is now described in the main text and in Fig 3. The log-log representation of somatic mutation rates vs lifespan or body size (Fig 3b-c) are also useful alternative representations of the data, as they clearly show that the allometric relationship between these variables holds across the full range of lifespans.

A complementary analysis that yields the same conclusion is to study the correlation between the end-of-lifespan mutation burden (ELB) and the lifespan of a species. Theory would predict that ELB should be similar across species and unrelated to the lifespan of a species. A regression of ELB vs lifespan confirms this (slope $P=0.39$).

2. Linear mixed effect regression models and bootstrapping

Our full dataset includes 208 crypts from 56 individuals from 16 species. When performing statistical analysis of the data, it is important to fully utilise the power of this design rather than collapsing the dataset into 16 data points. Linear mixed effect models enable us to perform robust regressions while accounting for the hierarchical structure of the data and the variance among individuals from each species and among crypts from each individual. These models strongly support that lifespan, rather than other life history variables, dominates the variation in somatic mutation rates across species. LME regression reveals that the fraction of variance explained (FVE, analogous to R^2) for $1/\text{lifespan}$ (1-parameter regression) is 0.82 ($P=2.9 \times 10^{-9}$). Using a 2-parameter

allometric regression, the FVE for $\log(\text{lifespan})$ is 0.85 ($P=1.0\times 10^{-6}$). In contrast, the FVEs for other life history variables alone are much lower: $\log_{10}(\text{body mass})$ (FVE=0.44, Fig 3f), litter size (FVE=0.51), $\log_{10}(\text{BMR})$ (FVE=0.27), BMR residuals (FVE=0.48); using 2-parameter LME models.

Having statistically shown that lifespan (and in particular the simpler $1/\text{lifespan}$ model) predicts somatic mutation rates much more strongly than any other life history trait studied, we then assessed whether other life history variables may still explain some variation in somatic mutation rates not already explained by lifespan. To do so, we used both nested LME models and partial correlation analyses. We fitted LME regression models using lifespan alone or lifespan in combination with one of the other life history variables. This analysis revealed that other life history variables had very little additional explanatory power (Fig 3f). We then used partial correlation analysis as an alternative approach to study this question, obtaining the same result (this is described in section 3 below and now shown in 2 new panels in Fig 3).

Following the reviewer's request, we have included these supplementary analyses in the main text. Given the hierarchical structure of the data, we think that the LME regression models, both with 1 and 2 variables, and the partial correlation analyses are superior to the use of subsampling or bootstrapping. However, motivated by the reviewer's question, we have also included a bootstrapping analysis in the manuscript (Methods). Two different bootstrapping tests were performed, resampling individuals with replacement or resampling species. These analyses demonstrate that the correlation between mutation rate and $1/\text{lifespan}$ is much more robust to bootstrapping at both the individual and species level than the correlation between mutation rate and $\log_{10}(\text{mass})$. Bootstrapping at the level of individuals yielded FVE estimates for lifespan and $\log_{10}(\text{mass})$ very similar to those using the full dataset (see blue curves in the image below). Resampling whole species also yielded supporting results, although as expected it caused considerable loss of information. The correlation with $1/\text{lifespan}$ remained strong across bootstraps, but the correlation with $\log(\text{body mass})$ varied in quality when resampling species. This was caused by the loss of informative species in some bootstraps. For example, naked mole-rat is a highly informative species in the present study (and in ageing research in general) for its long longevity despite its small body mass. Bootstrapping at the species level led to occasional resampled datasets without naked mole-rat, with a concomitant reduction in the power to distinguish the explanatory power of lifespan vs body mass. We have included these analyses at the

request of the reviewer, but the LME and partial correlation analyses are better suited to exploit the full dataset.

3. Partial correlation analysis

In addition to the nested LME regression models, we used partial correlation analysis to assess whether somatic mutation rates are associated with lifespan independently of body mass, and vice versa. Partial correlation measures the correlation of the residuals of two variables after removing the association of both variables with a third confounding variable (e.g. correlating the residuals of mutation rates and lifespan after removing their respective associations with body mass). This analysis clearly revealed that the association between somatic mutation rates and lifespan is independent of the body-mass variable, whereas the association between somatic mutation rates and body mass is lost after removing the association of both variables with lifespan (Fig 3b and 3d).

Importantly, this result remains after removing naked mole-rat from the data, showing that this claim is not dependent on the presence of naked mole-rat in the study. This new analysis without the naked mole-rat has been added to the Methods.

Extended Data Figure 14c. Distributions of regression FVE under bootstrapping. For the linear-mixed effects models regressing somatic mutation rates on inverse lifespan (no intercept; left) and log-mass (free intercept; right), each shaded curve presents the distributions of the fraction of inter-species variance explained (FVE) by each model, for 10,000 bootstrap replicates obtained through random resampling of either individuals (blue) or species (orange) (**Methods**). Vertical lines indicate the FVE values obtained using the entire data set.

Action: We have strengthened the statistical analyses of this part of the main text. We now describe the association between somatic mutation rates and lifespan using an unconstrained allometric model first (2 parameters), which justifies the use of the simpler and more meaningful 1/lifespan model (1 parameter). We have also added 4 new panels to Fig 3, including allometric (log-log) plots for lifespan and body mass, and the partial correlation allometric plots (formerly in Extended Data Figure 19). These analyses are now described in the main text, strongly supporting the original claims. We have also added the bootstrap analyses suggested by the reviewer (Methods section 16 and Extended Data Figure 14c).

Reviewer: 2. More philosophically speaking, is there a strong reason to believe that only one of the observed covariates (lifespan, body mass, BMR etc.) constitutes the “true” explanation for the mutation rate variation? Could they not all contribute, perhaps even to different degrees in the evolution of different species? Perhaps the authors can articulate their underlying thinking a bit more clearly on this point.

Authors’ response: We thank the reviewer for their question.

We would like to clarify that prior to performing this study, we did not expect a single variable to explain most of the variation of somatic mutation rates across species. However, this is what the data revealed. Three key analyses confirm that lifespan is by far the strongest prediction of the variation of somatic mutation rates:

1. LME regression models of each variable against somatic mutation rates reveals that lifespan has a much stronger correlation with somatic mutation rates than any other life history trait studied (Fig 3f).
2. LME regression models with lifespan, alone or in combination with other variables, revealed that other life history traits do not explain a significant amount of additional variance in somatic mutation rates. Likelihood ratio tests, $P=0.16$ for body mass, $P=0.92$ for litter size, $P=0.083$ for log-BMR, $P=0.79$ for allometric BMR residuals.
3. Finally, partial correlation tests were used to assess whether body mass or other life history variables could explain any of the variation in somatic mutation rates after removing the correlation with lifespan. These tests convincingly concluded that the association of somatic mutation rates with lifespan was strong

independently of body mass, whereas the correlation with body mass became non-significant after removing the correlation with lifespan.

In summary, in our dataset, 1/lifespan alone explained 82% of the variance in somatic mutation rates across mammalian species, with no other life history variable explaining a statistically significant fraction of the remaining variance. Whereas larger studies with many more species may identify small contributions from other variables, the overwhelming dominance of lifespan is highly unlikely to change given the results in our dataset.

Action: We have strengthened and clarified this section in the main text. We have also added two supplementary sections, on cancer risk and on ageing, that we hope help to contextualise and interpret these results.

Reviewer: 3. Can a similar analysis as described in point 1 be done for the zero intercept claim for those species where multiple individuals were sequenced (Figure 1c)?

Authors' response: We thank the reviewer for this suggestion.

We have attempted to do this following this request. Unfortunately, for most species the number of individuals is not large enough to allow a reliable bootstrap analysis. This is because the number of possible different bootstrap samples that can be produced from the data via resampling of individuals with replacement would be too small for accurate percentile confidence intervals to be inferred. For example, with 3 individuals, only 10 different bootstraps are possible (for further details about some of the problems resulting from bootstrap confidence intervals from small samples, see Hesterberg ¹). Despite these limitations, by restricting the analysis requested to those species with 5 or more individuals (>100 possible bootstraps per species), we can confirm that we obtain confidence intervals for the intercept overlapping zero: human (-1612, 1913), mouse (-83, 189) and rat (-3918, 117). Therefore, the bootstrapping results are consistent with those presented in the manuscript using linear regression models (Fig 1c and Supplementary Table 3), which we were able to run on all species with at least 3 individuals.

For completion, we have also run the LME regression model shown in Fig 3d using free intercepts; being a mixed-effects model, this takes advantage of the availability of multiple crypts per individual. Comparing the fit of this free-intercept LME model to the zero-intercept model also revealed a non-significant contribution of the intercepts, based on a Likelihood Ratio Test ($P=0.23$) and both the BIC and AIC model selection criteria (Methods, section 17). This shows that a zero-intercept LME model is more appropriate than a free-intercept model for the regression in Fig 3d.

Reviewer: 4. Can confidence intervals be added to the bars in Fig 3d, e.g. by bootstrapping?

Authors' response: We thank the reviewer for this useful suggestion. In response to this, we have performed a bootstrap analysis at the individual level (with the same method described in our answer to Question 1) to produce 95% bootstrap intervals for all the FVE estimates presented in Fig 3f (formerly 3d). We agree that this addition will help the readers to appreciate the robustness of our regression results.

Action: We have added 95% bootstrap intervals to the results presented in Fig 3f.

Reviewer: 5. It is interesting that rat and mouse have similar SBS1 rates, similar sizes, and similar lifespans, and yet rat has the most extreme SBSB rate, and mouse the most extreme SBSC rate. Can the authors hypothesize what might drive this difference? What mechanistic hypotheses could explain the different contributions of SBS1, SBSB and SBSC across the different species?

Authors' response: We thank the reviewer for this comment. This is indeed an interesting observation. It is difficult to speculate about the mechanistic bases for these differences across species, as we know little about what factors control the rates of these signatures even in human tissues. SBS1 is believed to be caused by 5-methylcytosine (5meC) deamination and repaired largely by mismatch repair. The aetiology of SBS5 remains largely unknown, although it may reflect the action of several forms of DNA damage followed by DNA repair independent of cell division². SBSC resembles signatures SBS18/36, believed to be associated with oxidative damage and whose repair is believed to be dominated by base excision repair.

In theory, differences in the rates of SBS1, SBSB and SBSC mutations could be explained by: (1) different rates of damage, (2) different rates of DNA repair, and/or (3) different DNA repair accuracies. For example, the rate of SBS1 could be affected by different levels of methylation (i.e. different frequencies of 5meC), different rates of spontaneous deamination, different activities of mismatch repair, different rates of cell division (replication can fix T:G mismatches caused by 5meC deamination before they are repaired) or even by different repair accuracies (MMR tends to repair T:G mismatches correctly to C:G, but accuracy could vary). Recent studies in individuals carrying mutations in the mismatch repair genes *MSH2/MSH6* or in the base excision repair gene *MUTYH* have reported dramatic increases in the rates of SBS1 and SBS18/36, respectively^{3,4}. This shows that, under normal circumstances, the vast majority of DNA damage events are correctly repaired, and that large changes in somatic mutation rates could be achieved through evolutionary pressure on key DNA repair enzymes.

At present, knowledge of all of the factors involved in the modulation of SBS1, SBSB and SBSC across species is too limited to speculate. But we hope that this study, as well as recent technical advances (e.g. Abascal et al. 2021), will motivate and facilitate future mechanistic studies on these questions.

Reviewer: 6. There are two entries for Human in ED Table 7.

Authors' response: We thank the reviewer for highlighting this. The two entries for human in ED Table 7 represent two different estimates for human intestinal stem cell dynamics obtained from the literature. We included them as separate entries to show the diversity of estimates and resulting uncertainty in these data.

Action: We have updated the table legend to clarify that the entries represent separate cell division estimates obtained from the literature.

Reviewer: 7. I apologize if I missed this in the methods, but are the authors worried that mutation calling in samples that are not entirely monoclonal (e.g. harbor porpoise, some of the horse and tiger samples) might be problematic? Might we be overestimating mutation burden by counting mutations from multiple crypts or missing some due to reduced sensitivity?

Authors' response: The reviewer is correct that polyclonal samples, if not handled correctly, can complicate the estimation of mutation burdens. For this reason, in this study we used a stringent filtering strategy to identify and exclude samples where polyclonality might lead to erroneous mutation rate estimates. Briefly, we used a truncated binomial mixture model to infer the clonal structure of each crypt. Only crypts composed of a single clone, and with sufficient coverage to ensure good mutation detection sensitivity, were used in the study. A detailed description of this process can be found in Methods (section 7) and is reproduced below (*italics added*).

“Our method for estimation of mutation rates assumes monoclonality of colorectal crypt samples. This assumption can be violated due to several causes, including contamination from other colorectal crypts during microdissection or library preparation, contamination with non-epithelial cells located in or near the crypt, insufficient time for a stem cell to drift to clonality within the crypt, or the possibility that in some species, unlike in humans, polyclonal crypts are the norm. Therefore, a truncated binomial mixture model was applied in order to remove crypts that showed evidence of polyclonality, or for which the possibility of polyclonality could not be excluded. An expectation–maximization (EM) algorithm was employed to determine the optimal number of variant allele fraction (VAF) clusters within each crypt sample, as well as each cluster’s location and relative contribution to the overall VAF distribution. ... After applying this model to the somatic substitutions identified in each sample, sample filtering was performed on the basis of the following three criteria.

Low mutation burden. We discarded samples presenting fewer than 50 somatic variants, which was indicative of low DNA quality or sequencing issues.

High mutation burden. We discarded samples with a number of somatic variants greater than three times the median burden of samples from the same individual (excluding samples with less than 50 variants). This served to exclude a small minority of samples presenting evident sequencing quality problems (such as low sequencing coverage), but which did not fulfill the low-VAF criterion for exclusion (see below).

Low VAF. We discarded samples in which less than 70% of the somatic variants were assigned to clusters with $VAF \geq 0.3$. However, this rule was not applied to those cases in which all the samples from the same individual had primary clusters with mean $VAF < 0.3$; this was done to prevent the removal of samples from individuals presenting high fractions of non-epithelial cells, but whose crypts were nonetheless dominated by a single clone.

These criteria led to the exclusion of 41 out of 249 samples. On the basis of visual assessment of sequencing coverage and VAF distributions, we decided to preserve three samples (ND0003c_lo0004, ND0003c_lo0011, TIGRD0001b_lo0010) which we considered to be clonal, but which would have been discarded based on the criteria above.”

This sample filtering strategy enabled us to identify and remove polyclonal samples, avoiding the overestimation of burden from samples with multiple crypts, or missing variants due to reduced sensitivity. As for those samples showing skewed VAF distributions in Extended Data Figure 3, these were either (i) samples belonging to individuals for which all samples had similarly low VAFs (see “Low VAF” filter above), which we therefore believed to be affected by contamination with non-crypt DNA, rather than polyclonality; or (ii) three samples that we chose to preserve in spite of our filtering (see last sentence of the quoted section), due to their showing qualitative evidence of clonality (e.g. mutation burdens comparable to those of clonal samples from the same individual). Hence, we are confident that these samples are not biasing our mutation burden estimates.

Reviewer: 8. The discussion might benefit from including some more controversial points. Currently the narrative is almost a little bit too “optimized” and smooth (at least for my taste). In their discussion, the authors write “The most striking finding of this study is the inverse scaling of somatic mutation rates with lifespan. This has been a long-standing prediction of the somatic mutation theory of ageing. The observation is consistent with a causative role of somatic mutations in mammalian ageing.” But some of the authors of this paper have recently published two Biorxiv preprints showing that individuals with increased mutation rates due to deficient polymerases or DNA mismatch repair do not show accelerated aging (only an increased risk of cancer). From the abstract of one of those Biorxiv papers: “The results, therefore, do not support a simple model in which all features of ageing are attributable to widespread cell malfunction directly resulting from somatic mutation burdens accrued during life.” How do the authors reconcile these findings with their speculation that mutation rates are a primary cause of aging?

Authors’ response: We thank the reviewer for this comment, which we hope has enabled us to strengthen this part of the manuscript. We originally found it difficult to include a full and detailed discussion of the nuances of different interpretations within the discussion section, given the stringent length constraints. Instead, we chose to give prominence to the new data in this study and write a shorter discussion listing different possible interpretations of the results, including both causal and non-causal interpretations.

However, motivated by this question, we have now written a new supplementary section to help contextualise our results within the large body of previous work on evolutionary and mechanistic theories of ageing. This section also discusses our results in the context of the two preprints mentioned by the reviewer. We point to this new section clearly in the main text, and hope that readers will find it helpful.

Action: We have added a new supplementary section to address these points.

Reviewer: 9. Also in the discussion, can the authors speculate on how their findings are expected to relate to cancer risk? Can they include some thoughts about the number of cells that undergo the described mutagenesis in each species and how this number may relate to organism-level cancer risk? Peto's paradox refers partially to lack of association between the number of cells in an organism and cancer risk. Even if individual end-of-life cells across species have a fairly similar mutation burden, overall cancer risk might still be expected to scale with the number of cells in an organism. Of course, we know that it does not, but I think it would be important to clarify that this work does not "solve" Peto's paradox (nor does it need to – that would be a very high bar).

Authors' response: We thank the reviewer for their valuable suggestion. We agree that the lack of a strong correlation between body mass and mutation rate suggests that species are likely to have evolved other mechanisms to "solve" Peto's paradox. To explore this question further, we have added a new supplementary section that uses a simple model of cancer risk (Armitage–Doll multistep model of carcinogenesis) to explain how large changes in cancer risk can be achieved with and without changes in the somatic mutation rate.

Action: We have added a new supplementary section (Supplementary Note 2) to expand on how our results relate to cancer risk.

Referee #2 (Remarks to the Author):

Reviewer: Cagan et al. asked whether somatic mutation rates scale across a selection of mammals in relation to the measured species-specific lifespan (80% max scored survivorship). Remarkably, they find that species-specific mutation rate inversely correlates with species-specific lifespan (short-lived species have higher mutation rates)

and that the life-long cumulative mutation burden is comparable across species. Their analysis is rigorous, convincing and their conclusions are fully supported by their results. The authors measured somatic mutation rates in intestinal crypts across a spectrum of mammals, which differ in body size, lifespan, metabolic rates and phylogenetic relatedness. They call somatic variants in intestinal crypts against skin, using a sequencing depth of 30x. They compute single-base-substitutions and indels in nuclear and mitochondrial genomes across all species. I congratulate the authors for a great piece of work. I greatly enjoyed reading this manuscript, which I consider a milestone and a breath of pure oxygen in the field of comparative genomics of aging.

What this work shows is a scale-free relationship between the rate of somatic mutations and species-specific lifespan. While the mechanistic part related with the mutational spectra across species alludes to specific shared mechanisms responsible for mutation rates to scale across species with lifespan - and this could have deserved further investigation and mechanistic analysis - the novelty and solidity of the main finding in my opinion is worthy of publication on nature.

Authors' response: We are grateful for the positive summary of our study.

Questions and comments:

Reviewer: 1. Overall, the quality of the figures is not "impressive" and rather cheap. I find that more effort could be placed in making figures 1b and 3a more intuitive to the readers.

Authors' response: We aimed to use simple representations to make them as clear as possible, but we appreciate the reviewer's feedback. In response to it, we have modified several panels to make them more intuitive to readers. The changes include the use of an improved colour scheme for Figs 1b and 3a, as well as additional colouring for panels Fig 3b-f. We have also added new panels to Fig 3.

Action: We have updated Figures 1 and 3 in an effort to make them more intuitive to readers.

Reviewer: 2. Have the authors "controlled" somatic mutation rate by estimated genome size (e.g. considered as an explanatory variable)? In several species, (germline, not somatic) mutation rate significantly scale with genome size (e.g., check Cui et al. <https://doi.org/10.1016/j.cell.2019.06.004> [doi.org]).

Authors' response: We thank the reviewer for this question. For the mammals included in this study the variation in genome size was very modest, with only a 1.3-fold difference between the smallest and largest genome (Supplementary Table 4). Following the reviewer's suggestion, we have tested the effect of incorporating total genome size as an explanatory variable using a linear mixed-effects model ($\text{mutation rate} \sim 1 / \text{lifespan} + \text{genome size}$). We found this variable's effect to be non-significant (fixed effect = -5.9×10^{-8} , $P=0.098$). A likelihood ratio test of this model against a model without genome size also supported a non-significant effect ($P=0.088$).

More importantly, because of the small variation in genome size and the large variation in somatic mutation rates per year across the species studied here, the conclusions are analogous when using alternative estimates of the somatic mutation rate. A regression analysis employing mutation rate per megabase, or mutation rate per coding exome, yielded results comparable to those of the analysis of mutation rate per genome (see Extended Data Figure 12). These results indicate that the variation in mutation rate is not explained by differences in genome size for these species.

The limited variation in genome size among the species included in this data set means they are not ideal for studying the impact of genome size on mutation rate. We hope that future studies including species with more variation in their genome size, and the relative proportion of their coding to non-coding genome, will lead to a better understanding of the relationship between genome size and somatic mutation rate.

Reviewer: 3. How far in phylogeny does the relationship between mutation rate and lifespan scale? What about century-long lived whales? What about super long-lived bats? What about beyond mammals? It would be nice if the authors could at least comment on this.

Authors' response: We are also very interested in these questions. Unfortunately, despite several years of sample collection through multiple collaborators, the need to work with fresh colon tissue samples collected at autopsy from captive individuals of known ages made it challenging to obtain additional samples from longer-lived mammals, including whales and bats. This is however something that we continue to pursue.

It will be of particular interest to know to what extent the relationship between mutation rate and lifespan scales in species with either extremely short or extremely long lifespans and, as the reviewer mentions, beyond mammals. We are currently in the process of collecting samples to address these questions. We are also developing more advanced sequencing methods that will not require autopsy samples, enabling studies even outside of vertebrates. However, all of this work is at a very early stage. We finish the manuscript acknowledging that similar studies across the wider tree of life will be of great interest to shed light on these questions, but at present we feel it would be premature to speculate on this within the manuscript without relevant data.

Reviewer: 4. I understand that the intercept for the mutation rate model is negligible, as represented in figure 1c. This finding justifies the use of a zero-intercept mixed effect model. However, it would be important to know what happens with a multilevel mixed effect model, where intercept and slope are "learned" from the data. This way, the authors can report an actual mean value for the intercepts in their model.

Authors' response: We thank the reviewer for their careful consideration of our mutation rate model. We may have misunderstood the question, but we believe that the reviewer is suggesting that we try to use free-intercept regression models to obtain estimates for both the intercepts and slopes of the relationship of mutation burden vs age in each species. We have done this in two ways in the manuscript. First, the linear regressions shown in Fig 1c are unconstrained regressions with free intercept and slopes. The estimates for each species for the slope using a zero-intercept regression, and for the slope and intercept using an unconstrained regression, are provided in Supplementary Table 3. The latter could only be run on species with at least 3 individuals. As the reviewer notes, all of the unconstrained regressions yielded non-significant intercepts. We thus used a zero-intercept assumption in the LME model in Fig 3. The motivation for doing so was to obtain much more precise estimates of the mutation rate per year with few individuals per species.

For completion, we also ran a LME model analogous to that in Fig 3 using free intercepts, as suggested by the reviewer. A likelihood ratio test comparing this model to the simpler model with zero-intercept across species yielded a non-significant effect for the addition of the intercepts ($P=0.23$) (Methods, section 17). Consistent with this, the zero-intercept model yielded lower values for both the Bayesian information criterion and the Akaike information criterion (Methods, section 17).

Referee #3 (Remarks to the Author):

Reviewer: In their study, the authors report a detailed survey of somatic mutation rates in 16 species. I am not aware of a previous attempt to measure somatic mutation rates in so many species and with these methods or others that would provide such a detailed comparative portrait of somatic mutation. Their findings were that somatic mutation rate scales best with lifespan rather than mass or other life history traits and that mutational patterns are largely the same across these mammals. They also report that despite 30-fold range in lifespan, the end of life mutational burden only ranges about 3 fold across these species. I found the approach to be powerful and the conclusions to be impactful.

While these specific methods are not my specialty, the authors convincingly argue that their model (the intestinal crypt) and methods (resequencing of multiple samples and individuals with detailed filtering) are sound and ideal for these questions. Key diagnostic results are also in-line with previously reported conclusions from human studies.

The analyses were also sound in my view and yet simple enough to be general and robust and understood by most readers. For example, the amount of variance in mutation rate explained by inverse age was not subtle (82%) and this speaks to the strength of that conclusion.

I find the study to be exciting, clear, well written and well executed. Proper controls were performed and alternative explanations were explored. From my perspective, I did not find many concerns. I would like to see this manuscript published at Nature.

Authors' response: We are very grateful for the positive summary of our manuscript.

Minor Concern:

Reviewer: 1. It would help the reader to have more explanation of how mutational signatures are derived and what they represent, when they are first presented.

Action: We thank the reviewer for their comment. We have added further explanation about mutational signatures in the manuscript (section "*Similar mutational signatures across mammals*").

References

1. Hesterberg, T. Bootstrap. *Wiley Interdiscip. Rev. Comput. Stat.* **3**, 497–526 (2011).
2. Abascal, F. *et al.* Somatic mutation landscapes at single-molecule resolution. *Nature* (2021) doi:10.1038/s41586-021-03477-4.
3. Sanders, M. A. *et al.* Life without mismatch repair. *bioRxiv* 2021.04.14.437578 (2021) doi:10.1101/2021.04.14.437578.
4. Robinson, P. S. *et al.* Inherited MUTYH mutations cause elevated somatic mutation rates and distinctive mutational signatures in normal human cells. *bioRxiv* 2021.10.20.465093 (2021) doi:10.1101/2021.10.20.465093.

Reviewer Reports on the First Revision:

Referees' comments:

Referee #1 (Remarks to the Author):

I thank the authors for addressing all comments so thoughtfully. I really enjoyed reading the new discussion and supplement, in particular supplementary note 1. The new supplementary text allows the motivated reader (and hopefully there will be many of the kind) to descend more deeply into work and to start grappling with some of the more profound questions about evolution that are raised by the authors' results. The idea that clonal expansions in normal tissues (instead of the "simple" mutation burdens of individual cells) contribute to aging phenotypes is certainly intriguing. Congratulations on this very nice piece of work!

Referee #2 (Remarks to the Author):

I am very pleased with the revised version and with how the authors have addressed all my comments. I have no further remarks and I look forward to seeing this work published in Nature.